# Changes in global air pollutant emissions during the COVID-19 pandemic: a dataset for atmospheric modeling

Thierno Doumbia[1], Claire Granier[1,2], Nellie Elguindi[1], Idir Bouarar[3], Sabine Darras[4], Guy Brasseur[3,5], Benjamin Gaubert[5], Yiming Liu[6], Xiaoqin Shi[3], Trissevgeni Stavrakou[7], Simone Tilmes[5], Forrest Lacey[5], Adrien Deroubaix[3], Tao Wang[6]

[1]Laboratoire d'Aérologie, Université de Toulouse, CNRS/UPS, Toulouse, France
[2]NOAA Chemical Sciences Laboratory and CIRES/University of Colorado, Boulder, CO, USA
[3]Max-Planck Institute for Meteorology, Hamburg, Germany
[4]Observatoire Midi-Pyrénées, Toulouse, France
[5]Atmospheric Chemistry Observations and Modeling, National Center for Atmospheric Research, Boulder, CO, USA
[6]Department of Civil and Environmental Engineering, The Hong Kong Polytechnic University, Hong Kong, China
[7]Royal Belgian Royal Institute for Space Aeronomy, Brussels, Belgium

*Correspondence to*: Thierno Doumbia (thierno.doumbia@aero.obs-mip.fr)

Claire Granier (claire.granier@aero.obs-mip.fr)

**Abstract.** In order to fight the spread of the global COVID-19 pandemic, most of the world countries have taken control measures such as lockdowns during a few weeks to a few months. These lockdowns had significant impacts on economic and personal activities in many countries. Several studies using satellite and surface observations have reported important changes in the spatial and temporal distributions of atmospheric pollutants and greenhouse gases. Global and regional chemistry-transport model studies are being performed in order to analyze the impact of these lockdowns on the distribution of atmospheric compounds. These modeling studies aim at evaluating the impact of the regional lockdowns at the global scale. In order to provide input for the global and regional model simulations, a dataset providing adjustment factors (AFs) that can easily be applied to current global and regional emission inventories has been developed. This dataset provides, for the January-August 2020 period, gridded AFs at a 0.1x0.1 latitude/longitude degree resolution, on a daily or monthly basis for the transportation (road, air and ship traffic), power generation, industry and residential sectors. The quantification of AFs is based on activity data collected from different databases and previously published studies. A range of AFs is provided at each grid point for model sensitivity studies. The emission AFs developed in this study are applied to the CAMS global inventory (CAMS-GLOB-ANT_v4.2_R1.1), and the changes in emissions of the main pollutants are discussed for different regions of the world and the first six months of 2020. Maximum decreases in the total emissions are found in February in Eastern China, with an average reduction of 20-30 % in NOx, NMVOCs and $SO_2$ relative to the reference emissions. In the other regions, the maximum changes occur in April, with average reductions of 20-30 % for NOx, NMVOCs and CO in Europe and North America and larger decreases (30-50 %) in South America. In India and African regions, NOx and NMVOCs emissions are reduced on average by 15-30 %. For the others species, the maximum reductions are generally less than 15 %, except in South America, where large decreases in CO and BC are estimated. As discussed in the paper, reductions vary highly across regions and sectors, due to the differences in the duration of the lockdowns before partial or complete recovery.

The dataset providing a range of AFs (average and average ± standard deviation) is called CONFORM (COvid-19 adjustmeNt Factors fOR eMissions) (https://doi.org/10.25326/88). It is distributed by the Emissions of atmospheric Compounds and Compilation of Ancillary Data (ECCAD) database (https://eccad.aeris-data.fr/).

# 1 Introduction

The COVID-19 pandemic has triggered different measures in most world countries to protect citizens from the spread of the Severe Acute Respiratory Syndrome-Coronavirus 2 (SARS-CoV-

2). The first measures, which included strict lockdowns, started in China at the end of January 2020. As the pandemic was spreading all over the world, lockdowns or other measures were gradually implemented in Asia, Europe, Oceania, North and South America and Africa. The restriction during the different lockdowns have resulted in significant changes in the emissions of greenhouse gases (Le Quéré et al., 2020; Liu et al., 2020; Han et al., 2021; Zheng et al., 2020) and reactive air pollutants (Forster et al., 2020; Venter et al., 2020). The impact of the reductions on the emissions of primary pollutants was assessed in different regions based on surface observations (e.g. Shi and Brasseur, 2020; Lee et al., 2020; Kim et al., 2020) and satellite retrievals (e.g. Bauwens et al., 2020; Zhang et al., 2020; Diamond et al., 2020; Biswal et al., 2020). Modeling studies have been initiated to investigate the changes in the global and regional distributions of tropospheric chemical compounds during the pandemics (Gaubert et al., 2021; Venter et al., 2020; Xing et al., 2020; Keller et al., 2020; Barré et al., 2020). These modeling studies are based on estimates of emissions for primary species, and on consistent changes of the emissions during the COVID-19 lockdown periods.

Here we present a global gridded dataset of emission adjustment factors (AFs) at a 0.1°x0.1° resolution and on a daily or monthly basis based on available activity data. Recent studies (Le Quéré et al., 2020; Forster et al., 2020; Liu et al., 2020) have discussed the changes in emissions at the global scale for several chemical compounds. In this paper, we propose a dataset of AFs for the individual sectors related to transportation including road, air and ship traffic, industry, residential activities and power generation. The advantage of such a dataset, which provides AFs for assessments of the impact of COVID-19 restrictions on pollutants emitted into the atmosphere, is that it can be applied directly to most global and regional inventory used in chemistry-climate and transport models in a flexible way. The inventories commonly used in these models (EDGAR (Crippa et al., 2018), ECLIPSE (Klimont et al., 2017), CAMS (Granier et al., 2019), MEIC (Li et al., 2017), etc.) include the sectors already mentioned (industrial processes, road transportation, power generation, residential, aviation and shipping). The emissions of these sectors are developed on the basis of a combination of several sub-sectors that are not provided separately in the emission inventories.

Section 2 describes the general methodology used in the development of the AFs. The following sub-sections analyze the sectoral activity data used to determine these factors. The availability of activity data depends strongly on the regions under consideration, and this might often lead to large uncertainties. We are therefore providing for each location and sector an estimated average, low and high values of the AFs. In Section 3, we present and discuss the sectoral changes in emission AFs over selected regions as a function of time.

This paper discusses the dataset developed for the January-August 2020 period: the dataset is called CONFORM (Covid-19 adjustmeNt Factors fOR eMissions) and will be updated to account for the latest available activity data information. The impact of the AFs on the emissions

developed as part of the CAMS global anthropogenic emissions (Granier et al., 2019; Elguindi et al., 2020) are discussed for the NMVOC, CO, NOx, BC, OC and $SO_2$ species in Section 3.5.


## 2 Methodology

To quantify the amount of the corresponding global gridded daily/monthly changes, the adjustment factors (AFs) are determined following the general methodology schematized in Figure 1. The four different steps taken in our study to estimate the AFs are the following: 1) A data survey is first
carried out to collect activity data for the power generation, industrial processes, residential, road transportation, shipping and aviation sectors. These sectors correspond to those considered in many global inventories such as the CAMS global anthropogenic emission inventory (CAMS-GLOB-ANT_v4.2_R1.1, Granier et al., 2019). The activity data used to estimate the emission AFs are available from numerous sources for different time scales, depending on the geographical area.
It should be noted that, in several regions of the world, accurate and/or up-to-date data are not publicly available, which could lead to a significant source of uncertainty in estimating reduction rates. Table 1 indicates the availability of data for the different sectors. Some of these data are used to support our analysis and contribute to the estimation of uncertainties on the AFs; 2) The collected data are then analyzed and an intercomparison of the changes in activity data of datasets
providing similar or equivalent parameters is performed. The dataset that provides the most detailed information and which meets better our needs is then chosen. The non-gridded AFs are calculated for each country or state/province according to the availability of activity data, as detailed in the following sections ; 3) The gridded daily/monthly netcdf files developed for each sector are obtained by assigning, to each cell,  the value of the AFs in the whole country or
state/province level corresponding to this grid cell: this is based on the fact that the lockdowns and restrictions have been generally taken at national or state level. For several sectors (road transport, industry and residential) the AF at the country or state/province level represents an average value calculated from several individual cities or locations; and 4) Finally, a comparison of the AFs derived from this study with other published data is performed, together with an evaluation of their
impact on emissions, using the CAMS-GLOB-ANT_v4.2_R1.1 inventory as a basis.

The dataset discussed in this paper covers the period from 1 January to 31 August 2020. It will be regularly updated to account for the latest available activity data information.

The AFs are calculated as the ratio between the activity data for a given sector and day or month, and the median value of the activity data over the five week-period starting on 1 January 2020. This corresponds to the baseline used in Google mobility trends. The median value is calculated from 1 January to 4 February for almost all regions, except for China where the first lockdown started on 23 January 2020 in the Hubei province. The reference values for China's activity data
are taken as the median daily value of the first three weeks (i.e. from 1 to 21 January 2020). The same baseline is used for all sectors, except for those for which only monthly activity data are

available. For those sectors, we consider the value of January 2020 as the baseline. We have
chosen the reference year as 2020 (pre-COVID-19), because detailed activity data such as mobility
data are only publicly available for the year 2020.


## 2.1 Road transportation

Several studies related to the COVID-19 pandemic have used the Google COVID-19 Community
Mobility data (https://www.google.com/covid19/mobility/) (Forster et al., 2020; Guevara et al.,
2020). This dataset provides the number of visitors to specific locations based on mobile phone
locations (e.g. parks, grocery and pharmacy stores, workplaces, retail and recreation, train stations
and residential) every day relative to a baseline value. This baseline is calculated as the median
value over the five-week period from January 3$^{rd}$ to February 6$^{th}$ 2020. We use measurements up to
31 August 2020 for all countries for which data are available (about 133 countries). For the USA,
the analysis was performed for each state. Google daily data are provided at country/state level and
generally cover several local areas (i.e. subregion or city). For example, in the USA, Google spans
the 50 states and the District of Columbia, as well as several cities in each state. The AFs for the
road transportation sector are derived from the estimation of transit usage (i.e. public transportation
including train stations, bus and subways) made by Google. In order to make the calculated AFs
comparable with those derived using the other data sources considered in this study, the AFs for
the Google's categories are scaled to 1 using the following formula AF = 1 + Google/100, so that
their values are less than 1 for a reduction in activity and above 1 otherwise.

It should be noted that Google mobility data are not available for all countries (e.g. China and for
nearly two thirds of African countries). Other datasets are providing mobility trends, such as the
Apple mobility trend reports (https://covid19.apple.com/mobility) and the TomTom
(https://www.tomtom.com/en_gb/traffic-index/ranking/) traffic congestion index. Apple measures
the number of requests for directions scaled relative to 13 January 2020, while TomTom provides
the percentage of the differences in time spent on a trip compared to uncongested conditions.
Google provides a better spatial coverage than the other two datasets. In Africa for example, Apple
and TomTom data are available for 3 and 2 countries, respectively, while Google provides data for
26 African countries. In China, TomTom reports measurements only for some cities. Unlike
Google and Apple, TomTom markets its data, which are therefore not in open access. In order to
175    evaluate the Google dataset, a regional comparison with Apple mobility data was performed
(Supplementary Information, Figures S1-2). First, we looked separately at one Apple category
(driving measurements, generated by counting the number of requested directions in transit or
public transport on Apple applications) and three Google categories (retail/pharmacy, workplaces
and grocery shopping destinations) according to three regions (Europe, USA and the rest of the
180    world (ROW)). The comparisons of temporal series show that, in general, Apple driving displays
much larger variations than the Google mobility changes (Figure S1). However, the patterns of the
average values of the four categories show a significant decrease in Europe of up to 57 % for

Apple driving and of 57, 65 and 31 % for Google workplaces, retail and grocery, respectively. The corresponding values in the USA are 43, 16, 41 and 40 %, respectively, while those in the ROW are 61, 31, 60 and 48 %. A comparison between monthly Google's non-residential (i.e. combining grocery, workplaces, transit and retail) and Apple driving data, which can be considered as equivalent categories, displays high correlation coefficients varying from 0.79 to 0.97 depending on the region (Figure S2). The data of these two categories agree relatively well, particularly during the first months of the COVID-19 pandemic (February to May) when lockdowns were strict in most regions. The largest discrepancies are observed from June to August, especially in Europe and the USA. These differences are likely due to the fact that the two datasets do not represent the same parameters: Apple bases their values on the volume of directions requests on phone applications while Google uses mobile phone locations. These differences can be attributed to a combination of several factors including the spatial coverage, the mode and category of transportation considered, the location of the measurements within the country or state/province. The calculation methods are also very different from one dataset to another.

Comparisons of the changes in the ground transportation sector for the different mobility datasets have been performed in recent studies (Forster et al., 2020; Le Quéré et al., 2020). The correlation coefficients between Google and Apple transit categories calculated in our study are in line with the value of 0.8 reported by Forster et al. (2020), for the February to June period. Liu et al. (2020) show trends from both Google and TomTom mobility datasets with the same order of magnitude during the first quarter of 2020. This analysis highlights the importance in the choice of the data to be considered in the estimation of changes based on activity data, especially for the more recent months, as well as for the future, when the dataset will be extended. Based on this analysis, we have used the Google Mobility data for estimating the AFs in regions where data are available.

In China, the AFs for road transportation were calculated based on Baidu Migration Scale index (https://qianxi.baidu.com/, available for China only). These indices are aggregated from migration flows within China, based on the positioning requests on Baidu Map Services. The index indicates the ratio between the number of people traveling in a city and the population of this city. The data are available only from 1 January to 2 May 2020 and cover about 343 Chinese cities in all provinces. Baidu officially stopped updating the dataset on its platform on 8 May 2020. We compared the changes in the Baidu Migration Scale Index with the relative difference of TomTom congestion levels for the Beijing, Tianjin, Chongqing and Shanghai Chinese areas. TomTom archived data are not freely available, but they can be retrieved from published graphs (https://www.tomtom.com/en_gb/traffic-index/ranking) providing the weekly changes in 2020 relative to the same periods in 2019. The results show a strong similarity between changes given by these two datasets for the period covered by the Baidu dataset (January to April), with a correlation coefficient of 0.9 (Figure S3). For the period from May to August 2020, Liu et al. (2020) showed that the difference in estimated $CO_2$ emissions from road transport between 2020 and 2019 is on average about 4% for May-November, with larger values for some days. Based on these analyses, and in order to cover the whole period of our study (January to August), we assume

that changes in road traffic in China after May 2020 are relatively low and close to those before the spread of the COVID-19 virus. These comparisons show that the proposed method for calculating the AFs (i.e. ratio between the activity data and the median value of activity data over a defined reference period) is consistent with changes in 2020 relative to the same period in 2019.

## 2.2 Industrial processes

This sector includes industrial production processes such as manufactured products from fossil fuel combustion, and represents a significant part of the emission sources of atmospheric pollutants. The 2020 data concerning industrial production are, however, not publicly available for many countries/regions. We used the crude steel production from the world steel association (Table 1), which is provided on a monthly basis, to estimate the rate of change in the industrial sector. Due to the difficulties to access the daily data in this source category, we assumed that changes in Google's workplace measures, which represent the percentage of people travelling to/from their workplaces, are representative of changes in industrial activities during the lockdowns. To verify our hypothesis, we compared the calculated monthly average AFs based on Google's workplace data for selected countries with those derived from crude steel productions (Figure S4). The average change in the first eight months of 2020 relative to the same periods in 2019 in crude steel production for the 24 countries shown in Figure S4 is 17 %, while the corresponding value using Google's workplace measures is 27 %. This indicates a fair agreement between the two datasets. However, there are large differences in some countries between these data. For example, in Europe the greatest change in crude steel production is 24 % in comparison to 59 % estimated using Google's workplace category, indicating a high level of uncertainty in the AFs for the industry sector.

## 2.3 Power generation

The power sector emission AFs were estimated by compiling several sources such as the total electricity load from the ENTSO-E (European Network of Transmission System Operators for Electricity) transparent platform for the European Union and the United Kingdom, the regional electricity demand from the EIA (Energy Information Administration) for the United States, the daily reports of the electricity generation from fossil fuel including coal, lignite and gas, Naptha and Diesel provided by the POSOCO (Power System Operation Corporation) for India, the thermal electricity production from the ONS (Operator of the National Electricity System) for Brazil and the daily power generation provided by the United Power system of Russia. We also used the local electricity demand data in Singapore. For Canada, we assumed that the data in Ontario are representative of power generation trends (Table 1). It should be noted that we did not apply any temperature correction to the electricity load. Liu et al. (2020) indicated that the COVID-19 and related restrictions explain about 85 % of the reduction in the power sector from January to March

and only 15 % are attributed to the temperature effect. For the rest of the world where data on electricity demand or thermal production are not publicly available, we estimate the AFs based on the data published by Le Quéré et al. (2020).

**2.4 Air transportation and shipping**

    We have used the monthly data on air transportation published by the Knowledge Center on Migration and Demography (KCMD) Dynamic Data Hub (Table 1). In addition to the observed passenger volumes from around the world, the KCMD provides air traffic scenarios covering the
COVID-19 period. Five scenarios are provided (Iacus et al., 2020). We selected the scenario called EUROC-L (L-shaped version) based on the Eurocontrol air traffic data, which provides global up-to-date monthly average air volumes. To assess the representativeness of the KCMD estimates, we compared the EUROC-L scenario data with the number of international scheduled flights from 14 countries in 2020 provided by the OAG (Official Aviation Guide)
(https://www.oag.com/coronavirus-airline-schedules-data). The results indicated that, when considering the global average, trends from both datasets are relatively similar, with a significant decrease of up to 65 % for OAG and 80 % for KCDM in April, May and June compared to the reference value in January 2020. However, since July there has been a slow recovery towards the pre-lockdown values (Figure S5, Supplementary Information). EUROC-L seems to overestimate
the changes during the period when restrictions were the most severe, especially in China, Japan and South Korea. In contrast, there is an underestimation of trends since July 2020 in most of the countries as shown in Figure S5. A lockdown for the whole of China was declared on February 10, 2020 and the country had its air traffic heavily impacted on the following days. This disruption does not appear in the KCDM data. Based on this analysis, we conclude that the KCDM data can
only be used to supplement the OAG data. It should be noted that KCDM data have the advantage of being available for over 230 countries and are regularly updated. Strohmeier et al. (2021) proposed another set of data on a daily basis, crowdsourced from air traffic control data using the OpenSky Network. When compared monthly, the Strohmeier et al. (2021) data show similar trends as the Official Aviation Guide (OAG) measurements that we used in conjunction with the KCDM
data.

    Detailed data on 2020 international and national shipping are not publicly available yet. In this study, the changes in shipping activity are determined based on the weekly containership port calls during the 31 weeks of 2020 (covering the period from 1 January to 2 August) compared to the
same period in 2019 and reported by the United Nations Conference on Trade and Development (UNCTAD) (https://unctad.org/news/covid-19-shipping-data-hints-some-recovery-global-trade). UNCTAD provides official statistics from marine traffic for different regions: North America (Canada, Mexico, United States), East Coast of South America (Argentina, Brazil, Uruguay), Northern Europe (Belgium, Germany, Netherlands), Southern Europe (France, Italy, Spain),
Northern and Western Africa (Egypt, Morocco, Nigeria, Togo), Eastern and Southern Africa

(Kenya, Tanzania, South Africa), South Asia (Bangladesh, India, Pakistan, Sri Lanka), South East Asia (Indonesia, Malaysia, Singapore, Thailand), China and Hong Kong. Weekly values are directly extracted from graphs and monthly AFs are derived.

## 2.5 Residential sector

In this study, the AF calculations for the residential sector are performed using measurements from Google's residential category, which covers most of the countries in the world. Google's residential category represents the duration during which people are constrained to their home through lockdown. This duration represents the additional time that people spent in places of residence due to the restrictions. In the CAMS-GLOB-ANT inventory, the residential sector includes mainly emissions due to cooking, heating and auxiliary engines that primarily use biomass or fossil fuels in households. The emissions from this sector vary according to the geographical zone due to the difference in lifestyles. We consider that, when people stay at home, the impact on heating use is moderate:  heating systems generally continue to operate during working hours, at a lower intensity. The extra time spent at home contributes significantly to other domestic activities, namely cooking, heating water and activities using fossil fuels. Based on this analysis, we expect an increase in residential combustion during the lockdown period.
In China, due to the lack of data, we use residential emissions published by Le Quéré et al. (2020) which are based on electricity demand for the city of London for the first fourth months of 2020. For the rest of the study period, we assumed that there is no change in the AFs for the residential sector in China.

## 3 Results

### 3.1 Adjustment factors for the transportation sector
This section focuses on the changes in the transportation sector which include road transportation, air traffic and shipping.

### 3.1.1 Road transportation
The AFs related to road transportation provide significantly lower emission values, when compared to typical values before the restrictions, and present large regional variations. Figure 2 displays the averages AFs in Europe, USA, South America, China, Africa and the rest of the world since the beginning of 2020. The regional variations of AFs (light pink color) are determined as the standard deviation of individual values for all the countries in the region or from local measurements in the country or state/province. We also calculated the minimum and maximum values within specific regions from all the values calculated for each of the countries in this region. An average daily reduction of up to 60 % in mid-April is shown in most regions of the world. In China and Europe, the reductions reached a maximum in mid-February and late March, respectively. In China, the lockdowns started on 23 January and the period from 24 January to 2

February coincides with Chinese Spring Festival holidays during which business activities were reduced. It has been reported that short term emissions and atmospheric composition reductions associated with the Spring Festival in China were about 10 % (Lin et al., 2011; He et al., 2021; Sun et al., 2020; Zhang et al., 2020; Zhang et al., 2021). The 2020 reductions in emissions from road transport are significantly larger and peak during February 2020, with no rebound after the Chinese New Year holiday (e.g. Kraemer et al., 2020; Miyazaki et al., 2020). In the USA where the variability is largest, the average decrease reached 40 % in mid-April. In Africa, the largest decrease occurred at the same period as in Europe and the USA, but with values of about 50 %, while the average decrease was much larger in South America, reaching 70 %. These results are in agreement with the changes during the first quarter of 2020 reported by Le Quéré et al. (2020) and based on Apple, TomTom mobility trends and local traffic data for the USA provided by MS2 (Modern Transportation Analytics) Corporation. Based on the standard deviations calculated from the data for each region and the evaluation results, an uncertainty of ±10 to ±30 % is associated with the estimated AFs, depending on the region. Figure 2 also indicates that some regions recovered to the pre-COVID-19 situation more rapidly than others (e.g. the USA, the European countries and China), while most countries in South America, Africa and the rest of the world continue to be affected by lockdown measures at the end of the period discussed in this paper.

### 3.1.2 Air traffic and shipping

As discussed in Section 2, the changes in air traffic were calculated using the monthly global scheduled flights from OAG combined with the passenger volumes reported from KCMD. Figure 3a displays the monthly air traffic AFs with an average value as low as to 0.2 (a decline of 80 %) in most of countries in the world. This large decrease spreads over several months from April to June, while the aviation activity started to rebound in July but the activity remained below the pre-pandemic level.

As a consequence of the global lockdown measures, activities in shipping also declined. Due to the lack of up-to-date data, we assume that changes in container ship port calls, published on the UNCTAD website, are representative of trends in shipping activities. The average global change as well as the associated standard deviation and upper and lower limits values are represented in Figure 3b. The number of port calls by container ship globally decline from January to August to 7 ± 6 % relative to 2019. However, changes vary across regions. For example, we found a monthly reduction in North America and Europe up to 18 % and 20 % in June, respectively. These reductions are in the lower limit values of 20-30 % reported from the literature and based on forecast and published reports (Le Quéré et al., 2020; Liu et al., 2020).

### 3.2 Adjustment factors for the industrial sector

Our estimates of AFs for the industry sector are derived from the Google's workplace measures, except for China for which factors are calculated based on $CO_2$ emissions published by Liu et al.

2020. Results show that the level of activity in the industrial sector over the 214 considered countries started, on average, falling down in mid-March when most of these countries began to

take restrictive measures. The average AF reaches a maximum decrease (up to 40 %) in April relative to the reference period (i.e., pre-COVID-19) before increasing until May. It remained relatively stable (approximately 20 % reduction) from the beginning of June to August. These levels of change are in line with those reported in Forster et al. (2020) for the first half of 2020. As Figure 4 illustrates, the average AF displays rather similar patterns across the different regions

considered, except in China. Based on Google data, industrial operations were subject to an important decrease in almost all regions, with a maximum daily average AF value close to 0.4 (60 % reduction) in the European and South American countries and in many other countries, especially in late March and early April. However, contrarily to the others countries, European countries show a second maximum daily average reduction (AF = 0.75 or 25 %) in mid-August,

with a magnitude lower than the first peak. The impact of the lockdowns on the industrial activities is somewhat smaller in the USA, Africa and in the rest of the world, with a maximum reduction of about 30-40 % relative to the pre-COVID-19 pandemic values. In China, AFs fell in mid-February to their minimum average values of 0.60 (40 % decrease), but rapidly increased to a complete recovery at the beginning of March and exceeded the pre-pandemic level by an average of 25 %

from April onward. The uncertainty range associated with the emission AFs for the industrial sector is evaluated to ±20 to ±30 %, depending on the region. It is noteworthy that for almost all countries except China, the pre-pandemic level of industrial activity has not been reached, eight months after the beginning of the first lockdown announcements.


### 3.3 Adjustment factors for the residential sector
Contrarily to other sectors, the AFs for the residential sector estimated from Google's mobility data show an increase of 20 to 30 % at its maximum, during the peaks of the lockdowns, depending on the region (Figure 5). The average percentage increase is 30 % in South America, while the

maximum average AF value in China is less than 1.10 (i.e. about 10 % increase). As for the others activity sectors, there is a variability of 10 % in all regions, as shown by the standard deviation. This increase in the adjustment factors is mostly due to the fact that, in most countries, schools were closed and teleworking was widespread. As a result, most people in countries affected by strict lockdowns had to stay home most of the time. The impact of lockdowns on residential

emissions is quite uncertain, as all the data including fuel use in commercial and residential buildings necessary to quantify this impact are not yet available worldwide. Our study, based on Google's residential measures reflecting time spent by people at their home, leads to increased emissions from the residential sector. However, Liu et al. (2020) reported a global decrease of about -2 % during the first seven months with an uncertainty of about 40 % using fuel

consumption data in residential buildings from 2019 that was scaled to 2020 based on the population-weighted heating degree days variation. In this study, the estimated uncertainty range in the emission AFs for residential sector reached ±20 %. Our global estimate is consistent with an

average increase of around 5 % in activity data from the residential sector, mainly during the strict lockdown period, estimated in Le Quéré et al. (2020).


### 3.4 Adjustment factors for the power generation sector

The COVID-19 pandemic has implications on the share of energy use in industry, commercial and domestic operations. In order to quantify the impact of the lockdowns in this sector, we used the data of electricity demand. The global demand experienced a maximum average decrease of 20 %

(AF= 0.8) between late March and early April when the restrictions were most stringent in Europe, USA, Africa and many other countries (Figure 6). From mid-June, there is an increase in the demand for electricity consumption in the USA which reaches a maximum value of about 20 % at the end of July and beginning of August but seems to be followed by a slow decrease. This can possibly be explained by an important demand in the commercial and residential sectors. It should

be noted that most African countries did not implement strict lockdowns: as discussed in Section 2, activity data are available mostly for North African countries and for South Africa, which experienced severe restrictions. The maximum decline ranged from 20 to 30 % in South American countries and in China. As for the other sectors, the peak of the reduction factor for energy in China and South American countries happened at the end of February and early April,

respectively. Our results are in line with the average reduction of 20 % or more of electricity demand in mid-March in several countries, as reported in the 2020 IEA energy review report (IEA, 2020). We estimated an uncertainty of ±15 % for the power sector, in agreement with the average values of the standard deviations calculated over all regions.


### 3.5 Impact on surface emissions

The estimations of emission AFs for six economic activity sectors (road transportation, industry, power generation, residential, shipping and aviation) discussed in the previous sections are provided by the CONFORM dataset on a daily basis, except for aviation and shipping calculated

on a monthly basis, for the period from January to August 2020 at a 0.1°x0.1° grid resolution. This dataset was also used in the Gaubert et al. (2021) paper, which examined changes in secondary atmospheric pollutants during the 2020 COVID-19 Pandemic using the Community Atmosphere Model (CAM-Chem) at the global scale. More evaluations of the CONFORM dataset have been performed in analyses of simulated and observed (in-situ and satellite measurements) atmospheric

surface ozone, nitrogen dioxide, and oxygenated volatile organic compounds across China by Liu et al. (2021) and Stavrakou et al. (2021). These studies demonstrate how reductions in anthropogenic emissions of ozone precursors (NOx and VOCs) as proposed by the CONFORM dataset contributed to the observed changes in the regions affected by the COVID-19 lockdowns.

The impact of the AFs changes on the total emissions (sum of emissions from transportation (road and non-road traffic), industry, residential, power and shipping) has been analyzed for different compounds and selected regions, using the anthropogenic CAMS-GLOB-ANT_v4.2_R.1.1 emission inventory (Granier et al., 2019; Elguindi et al., 2020). This dataset provides daily

emissions of the main atmospheric compounds, including speciated volatile organic compounds at
a 0.1°x0.1° resolution, from 2000 to 2020. Version R.1 of the CAMS-GLOB-ANT_v4.2 dataset
incorporates the MEIC1.3 regional inventory for China described by Zheng et al. (2018).
The percentage of global change in emissions during the COVID-19 compared to the reference
2020 emissions is shown in Figure 7 for the main pollutants (NOx, CO, $SO_2$, BC, OC and
NMVOCs) from January to June 2020. The vertical bars indicate the estimated lower and upper
475 limits of the changes in emissions due to the restrictions. Figure 7 shows that the changes depend
on the chemical species, with decreases in monthly global emissions of 22 % (11-26 %) for
NMVOCs, 17 % (12-24 %) for NOx, 14 % (7-18 %) for CO, 10 % (6-18 %) for $SO_2$ and 9 % (3-
13 %) for BC in April 2020 compared to a non-COVID-19 scenario. The global changes in OC
emissions are different from the other species with an increase of 3 % (0-7 %). These results
reflect the differences in the contribution of each sector to the total emissions for the different
species as suggested in Figure S6 (Supplementary material), which shows the absolute change in
emission per sector for NOx and OC. Figure S6 indicates that the reductions in NOx emissions are
mainly driven by the changes in road transport and industry sectors while there is a large
contribution of the residential sector in the OC emissions.
Figure 8, which displays regional monthly changes in Eastern China (20°N-45°N, 80°E-125°E),
Europe (35°N-70°N, 20°W-20°E), North America (20°N-50°N,135°W-35°W), South America
(60°S-20°N, 90°S-35°S), India (05°N-30°N, 60°E-90°E) and Africa (40°S-30°N, 20°W-40°E),
also indicates large differences in the changes in emissions among the regions. In agreement with
the changes in the activity data previously shown, the reductions are highest in February in Eastern
China region for all species, except OC for which a relatively small increase is observed during
that month. The average monthly NOx emissions in Eastern China derived from this study
decreased in February 2020 by 29 % (24-37 %) compared to the reference emission scenario (i.e.
without COVID-19 effect). The NMVOCs emissions decrease significantly, by 22 % (15-29 %).
The decreases in the amount of $SO_2$ emissions are of the same order of magnitude. BC and CO
show maximum average reductions of 8 and 10 %, respectively in the Eastern China region, while
OC shows a slight increase of 1.6 % for February.

In the rest of the world, NOx emissions exhibit large decreases (13-42 %) during the strictest
shutdown period (i.e. in April) when almost all sectors of activity slowed down or stopped. In
Europe, the average reduction in NOx emissions is 25 % (20-35 %) in April. These values are in
the same order of magnitude as the mean change of -33 % reported in Guevara et al. (2020) for
Europe for the period from 23 March to 26 April 2020. For $SO_2$, the average reduction in Europe
calculated by Guevara et al. (2020) is in the same order of magnitude as our low estimation range
of 9-23 % (average value of 14 %) in April. However, the percentage decline in NMVOCs
emissions (21-40 % with an average of 34 % as for April) derived from our study is much higher
than the value from Guevara et al. (2020). Solvents and industrial processes are the main sectors
contributing to anthropogenic NMVOCs emissions: in our study, the changes in the solvents
sectors, for which no data are yet available, are assumed similar to the changes related to the
industrial sector. The way the decrease in solvent emissions are handled in the different studies

could explain the large differences in the changes in NMVOCs emissions in different regions. European CO, $SO_2$ and BC emissions during the lockdowns decreased by an average of 8, 14 and 18 % in April, respectively, relative to the reference emissions.

The results show that the lockdowns led to an average reduction in NOx emissions in the USA of 21 % (15-35 %) during April, while these values reach 43 % (30-53 %) in South American regions, 29 % (22-42 %) in India and 13 % (8-19 %) in Africa. For NMVOCs, the maximum average reduction is 33 % (21-43 %) in the USA in April, while these values reach 55 % (27-57 %) in South America, 14 % (2-19 %) in India and 19 % (1.4-21 %) in Africa. The largest variabilities in the different countries in Africa and in India are due to the large uncertainties associated with the activity data in these countries. The magnitude of the changes in total emissions are not homogenous within the same region and can be very different from one location to another. Figure 9 displays the spatial distribution of the absolute difference between the COVID-19 and the reference scenarios as well as the associated percentage changes in NOx emissions for April 2020, the strictest lockdown period in most countries in the world. During that month, substantial declines in almost all geographical areas and for most species are seen, with the main decreases happening in urban areas strongly affected by human activities (Figure 9a and Figure S7 in Supplementary material). The percentage changes (Figure 9b) reflect the regional average decreases shown in Figure 8, with, for example, reductions over China ranging from 10 to 20 % in April 2020. It can be noticed that the emission changes in China result mostly from the significant decrease of emissions in the densely populated and heavily industrialized North China Plain and megacities. As already mentioned, the largest decreases in large cities of Europe, USA and India occurred in April, and in February in China (Figure S8, Supplementary material). The changes in total emissions of each species are driven by changes in the predominant sectors contributing to the emissions of each species. The observed decrease over the oceans from the different figures is related to the reduction in shipping activities in response to the slowdown of the economy.

These results provide a global overview of the effect of the lockdowns in the emissions of different atmospheric compounds in different regions of the world.

## 4 Conclusions

The restrictions and lockdowns resulting from the COVID-19 pandemic since the end of January 2020 have had important social, economic and environmental consequences. In this study, we have provided an estimate of adjustment factors (AFs) to quantify the changes in global emissions of major atmospheric pollutants during the lockdown periods. This dataset can be easily applied to the emissions used in global and regional models to simulate the impacts of the reduced human activity on the atmospheric composition and climate. To this purpose, we analyzed activity data from various sectors representative of transportation (road, air and ship), industry, residential and power generation. The resulting dataset provides daily or monthly sectoral AFs on a 0.1°x0.1° resolution over the globe, and can be used to quantify regional patterns in the distributions of emissions during the COVID-19 pandemic. When applied to the CAMS-GLOB-ANT_v4.2_R1.1

emissions dataset, large changes are estimated, with maximum decreases in the total emissions in February in Eastern China, with an average reduction of 20-30 % in NOx, NMVOCs and $SO_2$ emissions. In other regions, the maximum changes occur in April, with average reductions of 20-30 % for NOx, NMVOCs and CO in Europe and North America and larger decreases (30-50 %) in
South America. In both India and Africa, NOx emissions decline by 10 to 40 % while NMVOCs emissions decrease by 2 to 20 %, which larger uncertainties compared to other regions. For the others species, the maximum average reductions are generally less than 15 %, except in the South American countries where large decreases in CO and BC are estimated. These changes in the total emissions are related to the different changes in the individual sectors. The patterns of the
reductions also show large differences at the regional level.

We acknowledge that the lack of data for some activity sectors in several regions and the absence of accurate information on all sectors might cause significant uncertainties in the estimation of the AFs. For that reason, besides average values, the dataset also includes low and high estimation of
the AFs for the period over January-August 2020. This should give an estimate of the uncertainties on the distribution of chemical species in models. The AFs obtained in this study will be extended at least until the end of the year 2020, and will be revised and updated as new or improved information on economic and mobility activity becomes available.

## Data availability

The average estimated daily/monthly gridded AFs for the sectors considered in this study, i.e. transportation including air traffic and shipping, industries, residential and power generation are
available as NetCDF files for the global domain at a resolution of 0.1°x0.1° resolution. A range of AF is provided at each grid point as the average ± standard deviation. The acronym of the dataset is CONFORM (Covid-19 adjustmeNt Factors fOR eMissions), and it is available at https://doi.org/10.25326/88. The files (which have been updated to December 2020) can be openly accessed through the Emissions of atmospheric Compounds and Compilation of Ancillary Data
(ECCAD) database with a login account (https://eccad.aeris-data.fr/). In the ECCAD database, the dataset can be directly accessed using the link: https://eccad.aeris-data.fr/essd-conform/.

**Author contributions.**
**Conceptualization:** Thierno Doumbia, Claire Granier
**Production of emissions:** Thierno Doumbia**,** Sabine Darras, Nellie Elguindi, Claire Granier
**Collection of datasets:** Thierno Doumbia, Yiming Liu, Xiaoqin Shi, Tao Wang
**Analysis of the AFs results**: Thierno Doumbia, Idir Bouarar, Simone Tilmes, Benjamin Gaubert,
Trissevgeni Stavrakou
**Test of the AFs in atmospheric models**: Benjamin Gaubert, Adrien Deroubaix, Forrest Lacey

**Writing -original draft:** Thierno Doumbia, Claire Granier, Guy Brasseur
**Writing -review and editing:** All authors.

**Competing interests.** The authors declare that they have no conflict of interest.

**Acknowledgments.** We acknowledge the support of the AQ-WATCH European project, a HORIZON 2020 Research and Innovation Action (GA 870301), the ESA ICOVAC project, and the SEEDS EU project. The CAMS-GLOB-ANT dataset has been developed with the support of
the CAMS (Copernicus Atmosphere Monitoring Service, https://atmosphere.copernicus.eu/), operated by the European Centre for Medium-Range Weather Forecasts on behalf of the European Commission as part of the Copernicus Programme. The ECCAD database (eccad.aeris-data.fr) is supported by the French AERIS data infrastructure (aeris-data.fr). This material is based upon work supported by the National Center for Atmospheric Research, which is a major facility
sponsored by the US National Science Foundation under cooperative agreement no. 1852977. T.W. and Y.L. acknowledge support by the Hong Kong Research Grants Council (T24-504/17-N and A-PolyU502/16).

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

# Tables

Table 1: Data sources of activity data used to estimate the emission AFs. In this table, Rest Of World (ROW) refers to all world countries except China.

| Sectors | Country/ Region | Data | Data sources |
|---|---|---|---|
| Road transport (TRO) | China | Baidu Migration Scale Index (daily) TomTom Congestion Index[1] (weekly) [1]TomTom congestion traffic index are used to evaluate Baidu migration scale index | China Data Lab, 2020, "Baidu Mobility Data", https://doi.org/10.7910/DVN/FAEZIO https://www.tomtom.com/en_gb/traffic-index/ranking |
| | Rest Of World (ROW) | Google's transit stations (daily) Apple mobility[2] (daily) [2]Apple mobility data are used to evaluate Google activity data | https://www.google.com/covid19/mobility/ https://covid19.apple.com/mobility |
| Residential (RES) | China | Emissions from residential sector (daily) | Le Quéré et al. (2020) |
| | ROW | Google's residential (daily) | https://www.google.com/covid19/mobility/ |
| Industrial processes (IND) | China | Coal consumption from the six main coal producers (daily) | Liu et al. (2020) |
| | ROW | Google's workplaces (daily) | https://www.google.com/covid19/mobility/ |
| | World (ROW + China) | Crude steel production[3] (monthly) [3]Monthly used to help in the analysis of the adjustment factors in the industrial sector, estimated from Google's workplaces category. | https://www.worldsteel.org/ |
| Power Generation (ENE) | India | Production of Coal, Lignite, and Gas Naphtha and Diesel (daily) | Power System Operation Corporation Limited (https://posoco.in/reports/daily-reports/) |
| | USA (regional data) | Regional electricity load (daily) | Energy Information Administration (EIA) (https://www.eia.gov/beta/states/states/ca/data/dashboard/electricity) |
| | Europe | Total electricity load (daily) | ENTSO-E Transparent platform (https://transparency.entsoe.eu/dashboard/) |
| | Brazil | Thermal Production (daily) | Operator of the National Electricity System (http://www.ons.org.br/Paginas/) |
| | Russia | Power Generation (daily) | United Power System of Russia (http://www.so-ups.ru/index.php) |
| | Singapore | Electricity demand (daily) | https://www.ema.gov.sg/Statistics.aspx |
| | Canada | Electricity demand Ontario (daily) | http://reports.ieso.ca/public/Demand/ |
| | Other regions (e.g. Asia, Africa) | Emissions from Power sector (daily) | Forster et al. (2020) |
| Air transportation (AVI) | China and ROW | Projection of air traffic volume (monthly) | Knowledge Center on Migration and Demography (KCMD) Dynamic Data Hub https://bluehub.jrc.ec.europa.eu/migration/app/index.html# |
| Shipping (SHP) | World | Container shipping (monthly) | United Nation Conference on Trade and Development (UNCTAD) https://unctad.org/news/covid-19-shipping-data-hints-some-recovery-global-trade |


# Figures

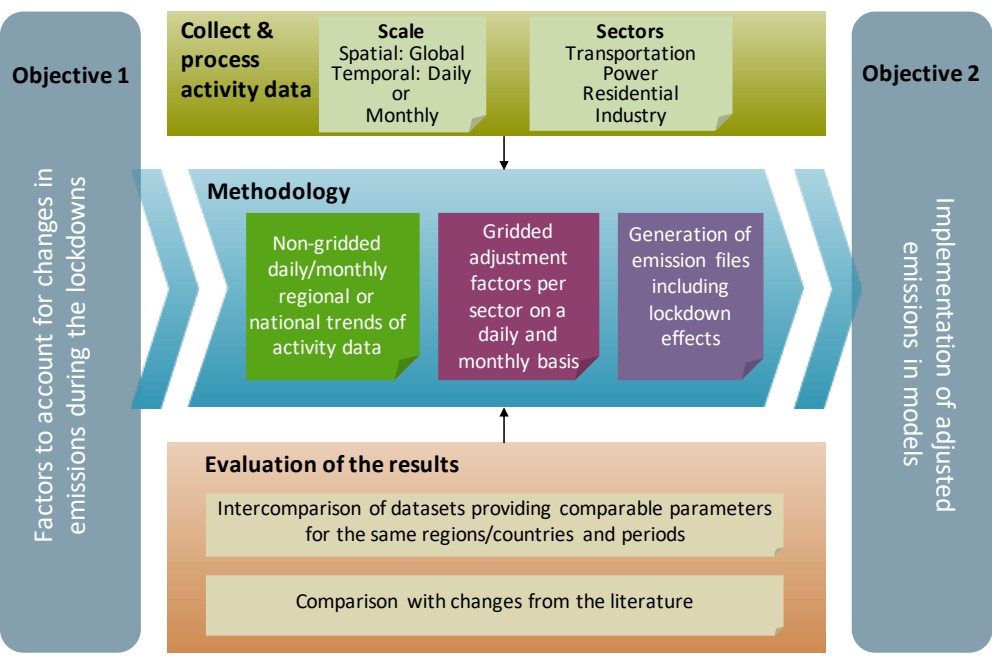

Figure 1: Schematic view of the different steps for estimating the emission adjustment factors (AFs).


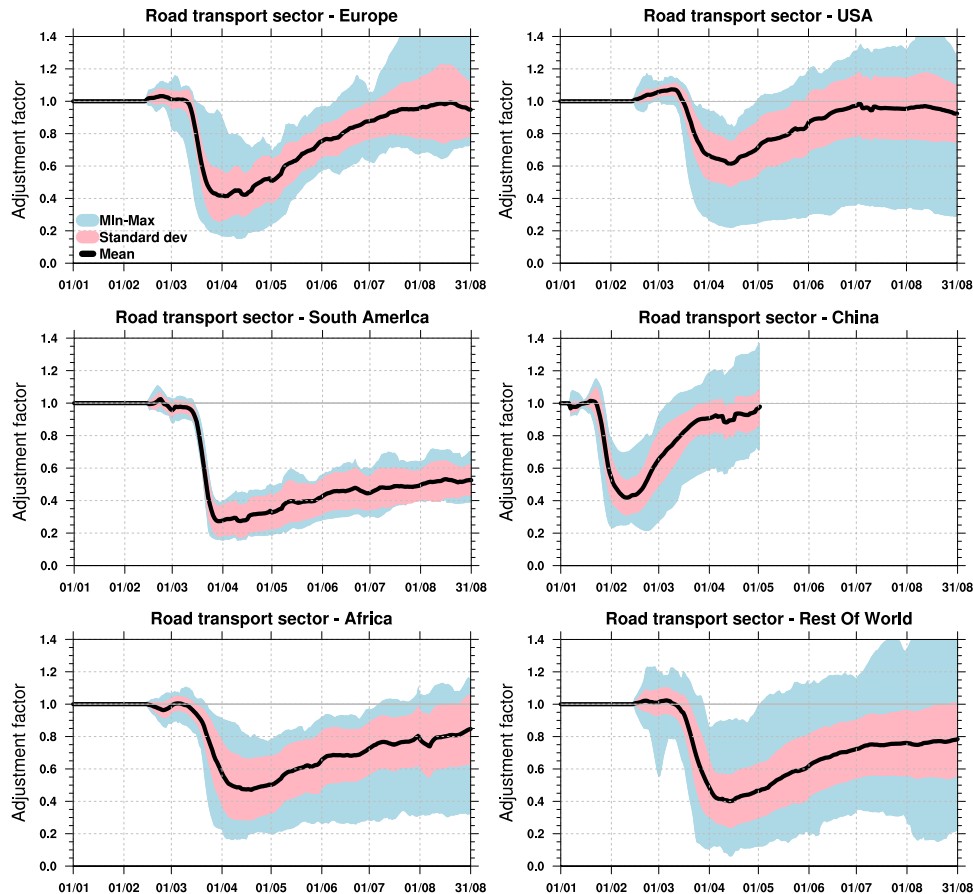

Figure 2: Daily AFs for the road transportation sector from January to August 2020 over Europe, USA, South America, China, Africa and the rest of the world. These estimations are derived from Google's transit category. The standard deviation values (light pink) result from the AFs for the individual countries or states/provinces. The light blue color indicates the range of the minimum and maximum values.

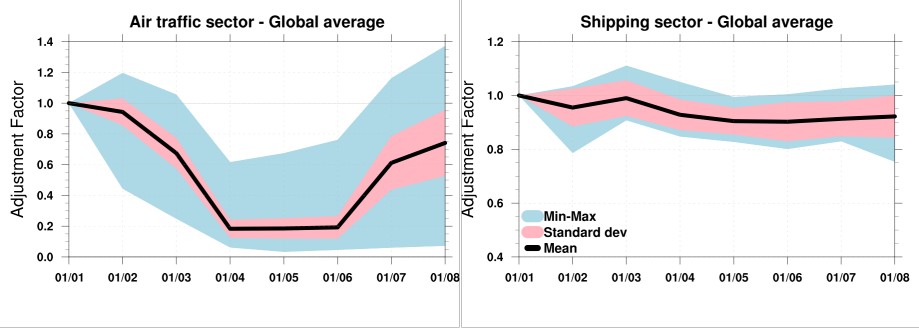

Figure 3: AF time series for air traffic and shipping as a function of month from January to August
2020. The standard deviation values (light pink) are calculated, based on the AFs of all individual
countries, while the light blue color indicates the range of the minimum and maximum values.

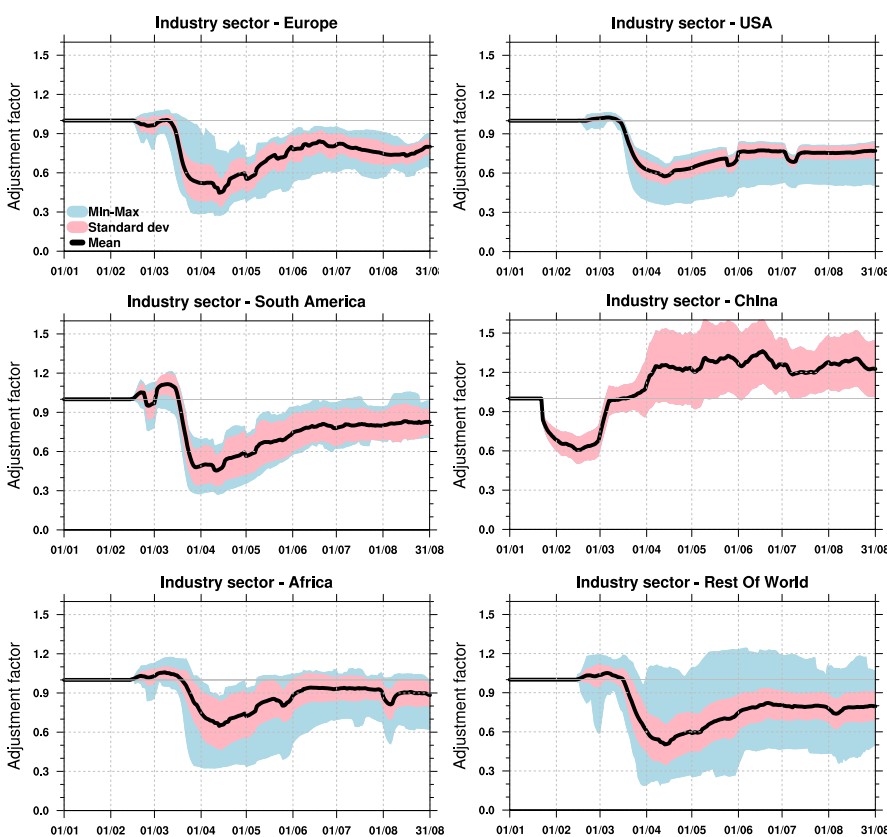

Figure 4: Daily AFs for the industrial sector from January to August 2020 over Europe, USA,
South America, China, Africa and the rest of the world. These estimations are derived from
Google's workplaces category. The standard deviation values (light pink) result from the AFs for
the individual countries or states/provinces. The light blue color indicates the range of the
minimum and maximum values.

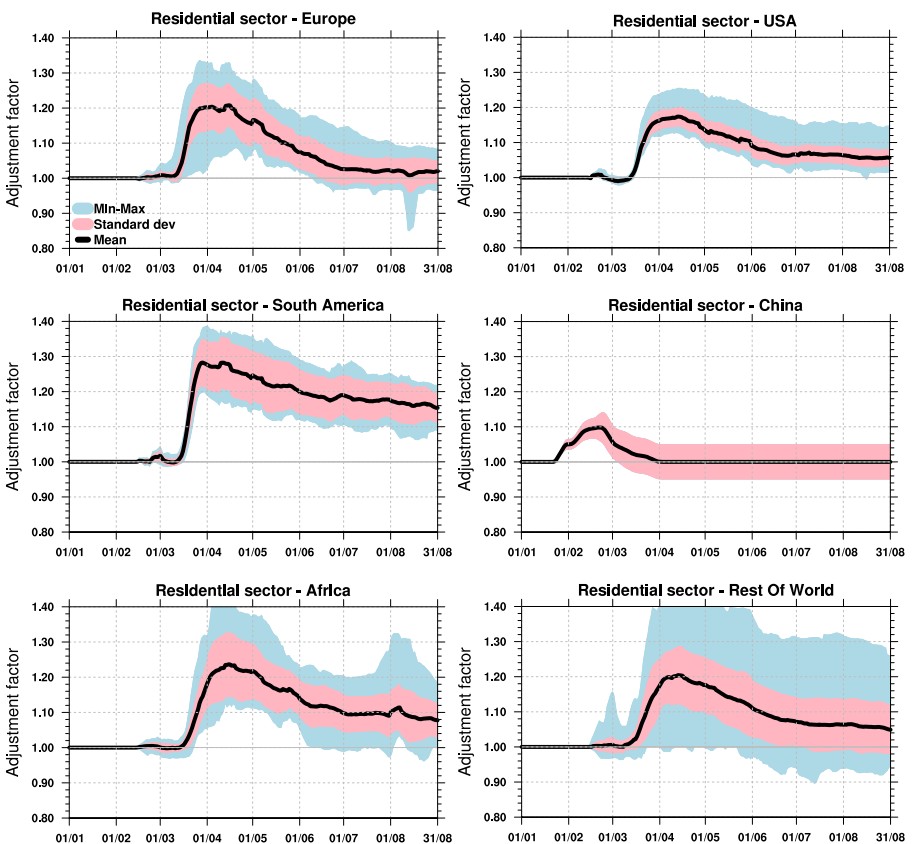


Figure 5: Daily AFs for residential sector from January to August 2020 over Europe, USA, South America, China, Africa and the rest of the world. These estimations are derived from Google's residential category. The standard deviation values (light pink) result from the AFs for the individual countries or states/provinces. The light blue color indicates the minimum and maximum

values.

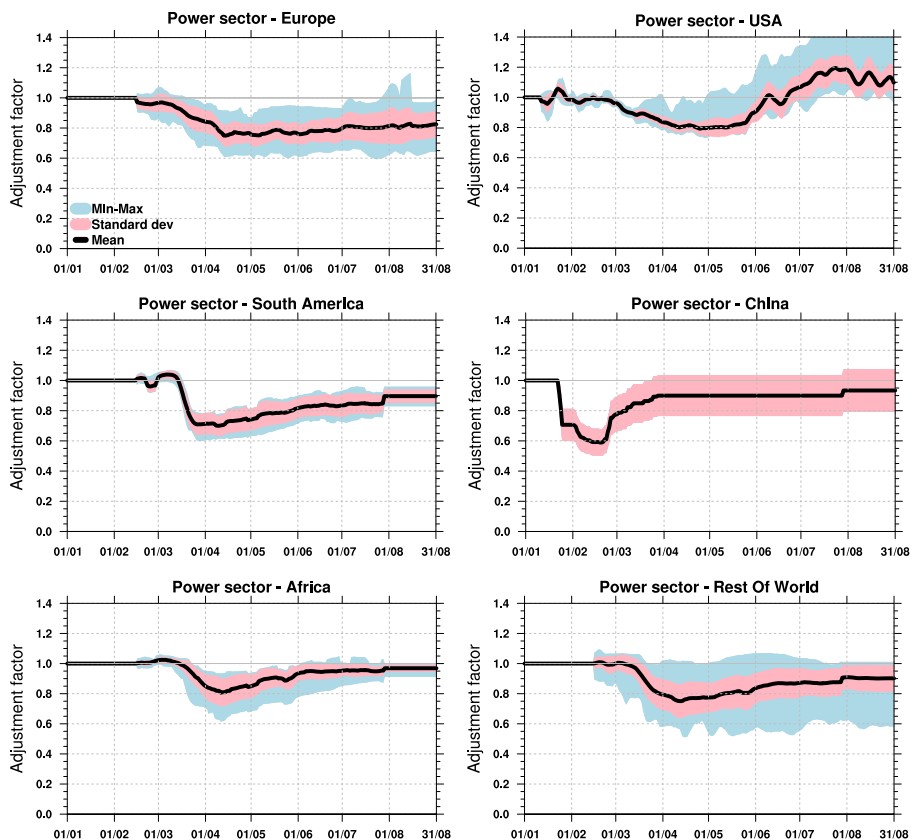


Figure 6: Daily AFs for power sector from January to August 2020 over Europe, USA, South America, China, Africa and the rest of the world. These estimations are derived from multiple data sources (Table 1). The standard deviation values (light pink) result from the AFs for the individual countries or states. The light blue color indicates the range of the minimum and maximum values.


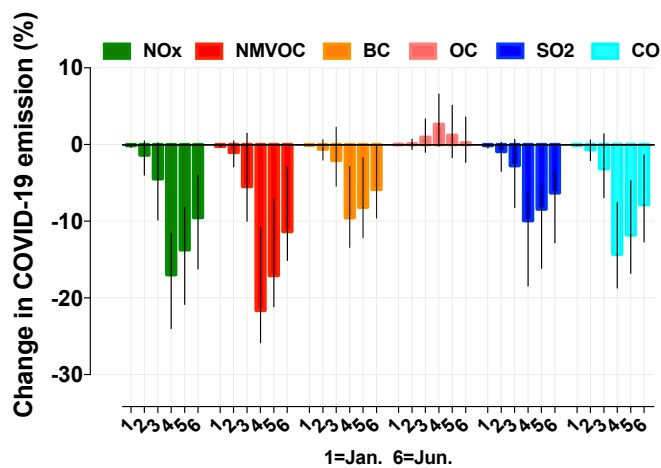

Figure 7: Global percentage change in total emissions (combination of emissions from ground transportation, industry, power, residential and shipping) of the main atmospheric compounds from January to June 2020. The vertical lines show the uncertainties associated to the estimated regional AFs.

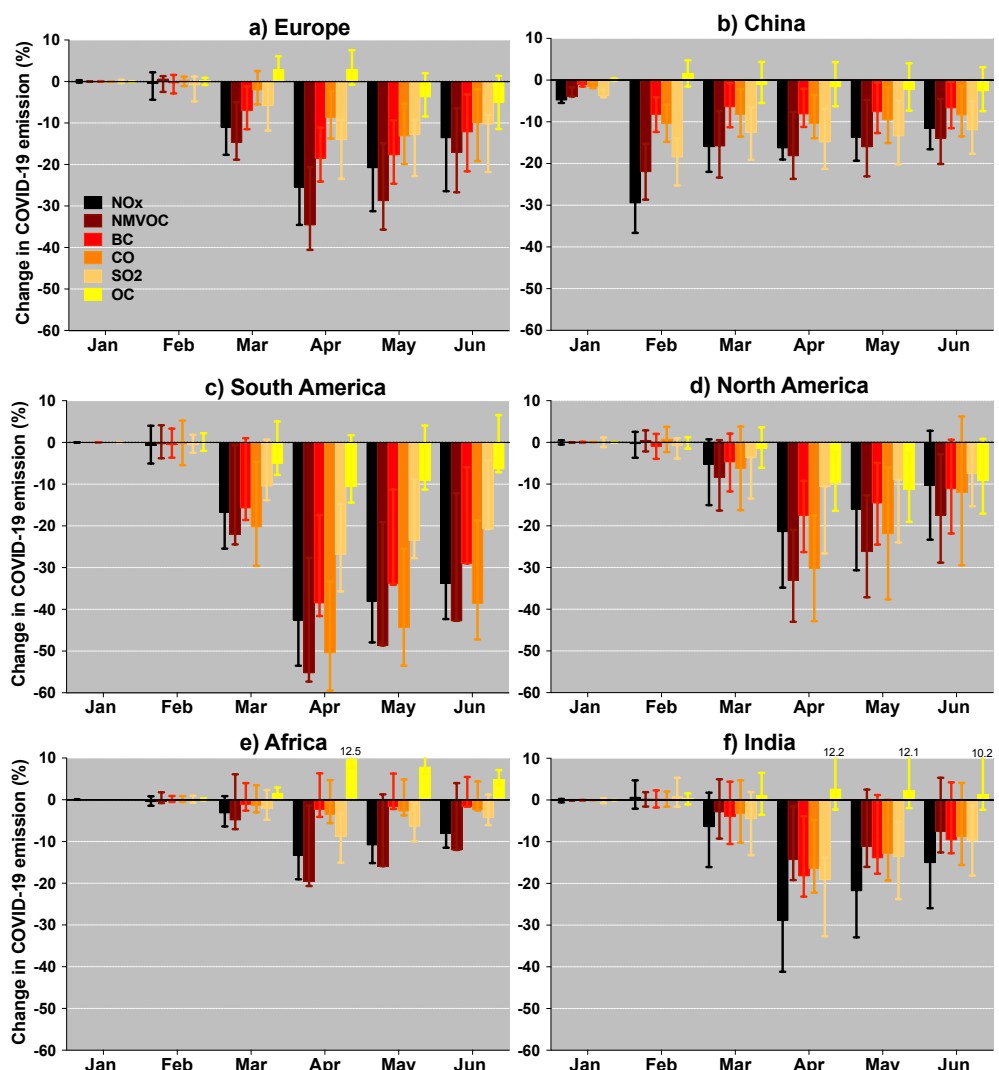

Figure 8: Percentage change in total emissions (combination of emissions from ground transportation, industry, power, residential and shipping), as a function of month for selected regions: a) Europe (35°N-70°N, 20°W-20°E), b) Eastern China (20°N-45°N, 80°E-125°E), c) South America (60°S-20°N, 90°S-35°S), d) North America (20°N-50°N,135°W-35°W), e) Africa (40°S-30°N, 20°W-40°E) and f) India (05°N-30°N, 60°E-90°E). The vertical lines show the uncertainties associated to the estimated regional AFs.

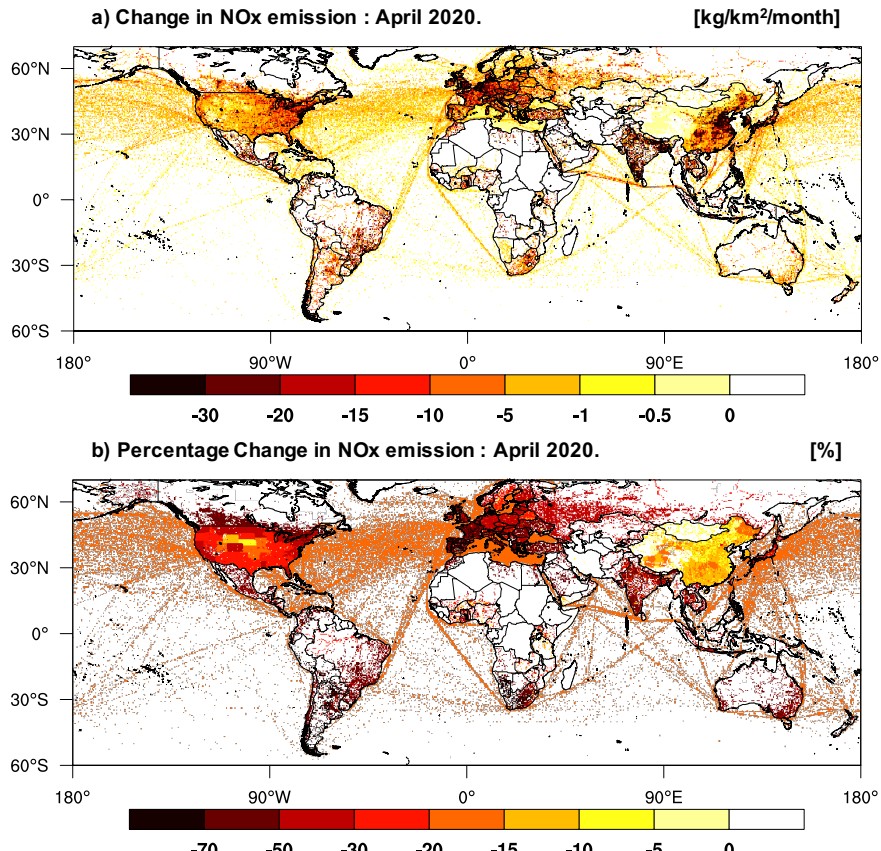

Figure 9: Spatial distribution of a) the absolute change and b) the percentage change in total NOx emission (combination of emissions from ground transportation, industry, power, residential and shipping) for April 2020.

875