# Peer review of "Changes in global air pollutant emissions during the COVID-19 pandemic: a dataset for atmospheric modeling"

_Earth System Science Data, 2020_

## Author Comment (AC1)

General comments:

Doumbia et al. present an interesting and important dataset on changes in global air pollutant emissions during the COVID-19 pandemic focusing on adjustment factors (AFs). This dataset is not only useful for global and regional emission inventories development, but also for atmospheric chemistry modeling, which is worth publishing in ESSD. I have several concerns on this paper.

1. Concerning the spatial distribution, the Methodology does not provide enough details on how the $0.1\times0.1$ dataset was created, how to support the $0.1\times0.1$ resolution? And the Results part showed only one spatial pattern figure (Fig. 8), which also need to be largely expanded not only for NOx.

2. Since the COVID-19 pandemic is still severe and the confirmed cases are increasing (https://www.worldometers.info/coronavirus/), this important dataset can aim to update it continuously, not only to the end of 2020 (lines 92-93).

3. How the AFs are calculated for different pollutants is not so clear with specific equations, especially when there are multiple sources activity data (e.g. industrial processes);

4. Why are there larger decreases (30-50 %) in South America than other areas (e.g. China, EU and US) for NOx, NMVOCs and CO. And Fig.8 seems to show that China has the largest decrease, rather than South America?

Some minor comments:

1. The title might be too limited for only "atmospheric chemistry modeling";

2. Line 51 are repeated for CO with lines 48-49;

3. Line 65 for greenhouse gas, there are other references (Han et al., 2021; Zheng et al., 2020);

4. Line 81-82, need to expand the meanings of this dataset? e.g. provide assessments on COVID-19 restrictions on pollutant emissions;

5. Line 95, could CO2 be included? It would be better for homology studies.

6. Lines 116-117: Add some introduction on CAMS -GLOB-ANT;

7. Lines 119-120 repeated with Lines 92-93;

8. Line 157: In reference to (Han et al., 2021), monthly road (and also ship) transportation data for China can be obtained at http://www.mot.gov.cn/tongjishuju/, but it needs some translation and digitalization work to obtain the data. And it is up to the authors to decide whether to include such data;

9. Lines 197-198: Maybe need some discussions. The assumption is not very consistent with Liu et al., 2020, see (https://www.carbonmonitor.org.cn/user/data.php?by=cn );

[Figure]

10. Lines 201-202: add a reference;

11. Line 205: May need to add time resolutions (e.g. daily or monthly) for sectors in Table 1;

12. Line 211: Cement is not included in this dataset?

13. Line 216: For China monthly data, the iron, steel and cement production can be obtained at https://data.stats.gov.cn/english/easyquery.htm?cn=A01 in "Output of major industrial products"

14. For Section 2.3, China's power data can be reflected by daily coal consumption at six main power groups, see (Han et al., 2021) Fig.4, and the data is provided at the end of this file.

15. Lines 251-252: Can compare with data from International Civil Aviation

Organization (ICAO). I noticed a report which contained regional/country data (https://www.icao.int/sustainability/Documents/COVID-19/ICAO_Coronavirus_ Econ_Impact.pdf ), which you may find useful in cross validation;

[Figure]

[Figure]

16. Line 324: Also consider these refs. (He et al., 2021; Sun et al., 2020; Zhang et al., 2021) ;

17. Lines 375-377: Maybe not this case (see below 3 figures), and CO2 emissions from Liu et al., (2020) (https://www.carbonmonitor.org.cn/user/data.php?by=cn) and NBS statistical data on iron and steel and cement productions (https://data.stats.gov.cn/english/easyquery.htm?cn=A01) showed that industry in China recovered soon after April 1[st] , and surpassed the before COVID-19 mean state.

[Figure]

[Figure]

[Figure]

18. Line 406: Should be Figure 6 for power and Figure 4 for Industry?

19. Lines 411-412: Power AFs for China is not consistent with (Liu et al., 2020) and (Han et al., 2021), I provided the daily coal consumption data for six major power generation groups at the end of minor comments for your reference.

20. Line 426: change the red color "is" to black;

21. Lines 449-450: NMVOCs are mainly from solvents and industrial processes (Lines 464-465), and not homogeneous with SO2? Here "rather similar" seems to show some relations?

22. Lines 478-480: Expand this short paragraph a bit;

23. Line 486: "largest" or "large" ?

24. Line 513: "might" or "could" or "did"?

25. Draw vertical lines to show lockdown and unlock date information in time series figures for major countries or regions.

Daily coal consumption at six main power generation groups from (Han et al., 2021). Data were derived from https://www.wind.com.cn/ .

| Date | Daily coal consumptions (10^4 ton) |
|---|---|
| 2019-12-01 | 73.07 |
| 2019-12-02 | 72.67 |
| 2019-12-03 | 69.55 |
| 2019-12-04 | 71.45 |
| 2019-12-05 | 73.03 |
| 2019-12-06 | 76.28 |
| 2019-12-07 | 76.40 |
| 2019-12-08 | 74.67 |
| 2019-12-09 | 75.11 |
| 2019-12-10 | 75.22 |
| 2019-12-11 | 73.95 |
| 2019-12-12 | 73.99 |
| 2019-12-13 | 74.44 |
| 2019-12-14 | 74.31 |
| 2019-12-15 | 71.13 |
| 2019-12-16 | 68.49 |
| 2019-12-17 | 67.36 |
| 2019-12-18 | 69.90 |
| 2019-12-19 | 71.79 |
| 2019-12-20 | 76.26 |
| 2019-12-21 | 76.66 |
| 2019-12-22 | 77.28 |
| 2019-12-23 | 76.12 |
| 2019-12-24 | 76.04 |
| 2019-12-25 | 77.10 |
| 2019-12-26 | 77.43 |
| 2019-12-27 | 77.24 |
| 2019-12-28 | 76.56 |
| 2019-12-29 | 75.27 |
| 2019-12-30 | 75.47 |
| 2019-12-31 | 77.17 |
| 2020-01-01 | 77.65 |

| | |
|---|---|
| 2020-01-02 | 73.98 |
| 2020-01-03 | 70.26 |
| 2020-01-04 | 75.40 |
| 2020-01-05 | 76.06 |
| 2020-01-06 | 74.17 |
| 2020-01-07 | 70.08 |
| 2020-01-08 | 70.10 |
| 2020-01-09 | 70.49 |
| 2020-01-10 | 71.79 |
| 2020-01-11 | 74.16 |
| 2020-01-12 | 72.90 |
| 2020-01-13 | 68.96 |
| 2020-01-14 | 66.56 |
| 2020-01-15 | 68.39 |
| 2020-01-16 | 67.83 |
| 2020-01-17 | 67.13 |
| 2020-01-18 | 65.65 |
| 2020-01-19 | 62.65 |
| 2020-01-20 | 59.18 |
| 2020-01-21 | 55.79 |
| 2020-01-22 | 51.01 |
| 2020-01-23 | 47.21 |
| 2020-01-24 | 46.93 |
| 2020-01-25 | 45.03 |
| 2020-01-26 | 43.53 |
| 2020-01-27 | 42.73 |
| 2020-01-28 | 41.13 |
| 2020-01-29 | 43.73 |
| 2020-01-30 | 42.83 |
| 2020-01-31 | 39.93 |
| 2020-02-01 | 38.51 |
| 2020-02-02 | 37.55 |
| 2020-02-03 | 38.05 |
| 2020-02-04 | 37.76 |
| 2020-02-05 | 37.66 |
| 2020-02-06 | 37.04 |
| 2020-02-07 | 37.42 |
| 2020-02-08 | 37.52 |
| 2020-02-09 | 37.82 |
| 2020-02-10 | 36.92 |
| 2020-02-11 | 37.20 |
| 2020-02-12 | 37.24 |
| 2020-02-13 | 38.49 |
| 2020-02-14 | 38.11 |
| 2020-02-15 | 38.00 |
| 2020-02-16 | 38.20 |
| 2020-02-17 | 38.80 |
| 2020-02-18 | 38.93 |
| 2020-02-19 | 39.33 |
| 2020-02-20 | 42.14 |
| 2020-02-21 | 42.13 |
| 2020-02-22 | 42.08 |
| 2020-02-23 | 41.88 |
| 2020-02-24 | 42.67 |

| | |
|---|---|
| 2020-02-25 | 41.57 |
| 2020-02-26 | 42.86 |
| 2020-02-27 | 42.75 |
| 2020-02-28 | 43.11 |
| 2020-02-29 | 43.36 |
| 2020-03-01 | 43.36 |
| 2020-03-02 | 45.07 |
| 2020-03-03 | 45.60 |
| 2020-03-04 | 47.68 |
| 2020-03-05 | 49.33 |
| 2020-03-06 | 50.41 |
| 2020-03-07 | 50.81 |
| 2020-03-08 | 52.41 |
| 2020-03-09 | 52.81 |
| 2020-03-10 | 51.56 |
| 2020-03-11 | 52.52 |
| 2020-03-12 | 50.73 |
| 2020-03-13 | 53.38 |
| 2020-03-14 | 53.47 |
| 2020-03-15 | 52.25 |
| 2020-03-16 | 54.17 |
| 2020-03-17 | 51.82 |
| 2020-03-18 | 54.66 |
| 2020-03-19 | 55.03 |
| 2020-03-20 | 54.57 |
| 2020-03-21 | 55.01 |
| 2020-03-22 | 54.77 |
| 2020-03-23 | 55.01 |
| 2020-03-24 | 57.44 |
| 2020-03-25 | 61.67 |
| 2020-03-26 | 58.68 |
| 2020-03-27 | 59.94 |
| 2020-03-28 | 56.44 |
| 2020-03-29 | 55.50 |
| 2020-03-30 | 55.75 |
| 2020-03-31 | 56.10 |
| 2020-04-01 | 57.44 |
| 2020-04-02 | 58.72 |
| 2020-04-03 | 56.53 |
| 2020-04-04 | 56.87 |
| 2020-04-05 | 54.55 |
| 2020-04-06 | 51.72 |
| 2020-04-07 | 52.05 |
| 2020-04-08 | 54.12 |
| 2020-04-09 | 54.08 |
| 2020-04-10 | 55.31 |
| 2020-04-11 | 56.11 |
| 2020-04-12 | 56.12 |
| 2020-04-13 | 53.66 |
| 2020-04-14 | 50.62 |
| 2020-04-15 | 54.82 |
| 2020-04-16 | 57.88 |
| 2020-04-17 | 56.42 |
| 2020-04-18 | 55.65 |

| | |
|---|---|
| 2020-04-19 | 56.95 |
| 2020-04-20 | 56.94 |
| 2020-04-21 | 55.91 |
| 2020-04-22 | 58.02 |
| 2020-04-23 | 57.06 |
| 2020-04-24 | 57.23 |
| 2020-04-25 | 55.98 |
| 2020-04-26 | 56.08 |
| 2020-04-27 | 55.01 |
| 2020-04-28 | 52.23 |
| 2020-04-29 | 53.60 |
| 2020-04-30 | 53.88 |
| 2020-05-01 | 53.19 |
| 2020-05-02 | 52.73 |
| 2020-05-03 | 48.74 |
| 2020-05-04 | 50.38 |
| 2020-05-05 | 54.89 |
| 2020-05-06 | 58.10 |
| 2020-05-07 | 59.92 |
| 2020-05-08 | 63.32 |
| 2020-05-09 | 63.24 |
| 2020-05-10 | 65.77 |
| 2020-05-11 | 65.94 |
| 2020-05-12 | 64.20 |
| 2020-05-13 | 66.47 |
| 2020-05-14 | 65.15 |
| 2020-05-15 | 65.99 |
| 2020-05-16 | 68.20 |
| 2020-05-17 | 69.74 |
| 2020-05-18 | 68.51 |
| 2020-05-19 | 69.18 |
| 2020-05-20 | 65.63 |
| 2020-05-21 | 67.84 |
| 2020-05-22 | 67.54 |
| 2020-05-23 | 66.72 |
| 2020-05-24 | 66.96 |
| 2020-05-25 | 65.34 |
| 2020-05-26 | 63.72 |
| 2020-05-27 | 66.23 |
| 2020-05-28 | 62.59 |
| 2020-05-29 | 59.30 |
| 2020-05-30 | 58.76 |
| 2020-05-31 | 58.93 |
| 2020-06-01 | 57.76 |
| 2020-06-02 | 55.79 |
| 2020-06-03 | 58.02 |
| 2020-06-04 | 61.62 |
| 2020-06-05 | 60.09 |
| 2020-06-06 | 63.04 |
| 2020-06-07 | 63.73 |
| 2020-06-08 | 60.30 |
| 2020-06-09 | 57.52 |
| 2020-06-10 | 61.82 |
| 2020-06-11 | 64.73 |

| | |
|---|---|
| 2020-06-12 | 65.96 |
| 2020-06-13 | 67.65 |
| 2020-06-14 | 67.98 |
| 2020-06-15 | 67.06 |
| 2020-06-16 | 66.59 |
| 2020-06-17 | 67.92 |
| 2020-06-18 | 65.22 |
| 2020-06-19 | 67.00 |
| 2020-06-20 | 66.32 |
| 2020-06-21 | 66.55 |
| 2020-06-22 | 63.10 |
| 2020-06-23 | 62.35 |
| 2020-06-24 | 63.59 |
| 2020-06-25 | 65.24 |
| 2020-06-26 | 62.48 |
| 2020-06-27 | 60.11 |
| 2020-06-28 | 61.29 |
| 2020-06-29 | 63.67 |
| 2020-06-30 | 63.84 |
| 2020-07-01 | 64.67 |
| 2020-07-02 | 62.52 |
| 2020-07-03 | 61.97 |
| 2020-07-04 | 62.57 |
| 2020-07-05 | 62.67 |
| 2020-07-06 | 62.27 |

**References:**

Han, P. F., Cai, Q. X., Oda, T., Zeng, N., Shan, Y. L., Lin, X. H., and Liu, D.: Assessing the recent impact of COVID-19 on carbon emissions from China using domestic economic data, Science of the Total Environment, 750, 2021.

He, Y., Pan, Y., Gu, M., Sun, Q., Zhang, Q., Zhang, R., and Wang, Y.: Changes of Ammonia Concentrations in Wintertime on the North China Plain from 2018 to 2020, Atmospheric Research, doi: https://doi.org/10.1016/j.atmosres.2021.105490, 2021. 105490, 2021.

Liu, Z., Ciais, P., Deng, Z., Lei, R., Davis, S. J., Feng, S., Zheng, B., Cui, D., Dou, X., Zhu, B., Guo, R., Ke, P., Sun, T., Lu, C., He, P., Wang, Y., Yue, X., Wang, Y., Lei, Y., Zhou, H., Cai, Z., Wu, Y., Guo, R., Han, T., Xue, J., Boucher, O., Boucher, E., Chevallier, F., Tanaka, K., Wei, Y., Zhong, H., Kang, C., Zhang, N., Chen, B., Xi, F., Liu, M., Bréon, F.-M., Lu, Y., Zhang, Q., Guan, D., Gong, P., Kammen, D. M., He, K., and Schellnhuber, H. J.: Near-real-time monitoring of global CO2 emissions reveals the effects of the COVID-19 pandemic, Nature Communications, 11, 5172, 2020.

Sun, Y., Lei, L., Zhou, W., Chen, C., He, Y., Sun, J., Li, Z., Xu, W., Wang, Q., Ji, D., Fu, P., Wang, Z., and Worsnop, R. D.: A chemical cocktail during the COVID-19 outbreak in Beijing, China: Insights from six-year aerosol particle composition measurements during the Chinese New Year holiday, ence of The Total Environment, 742, 2020.

Zhang, Q., Pan, Y., He, Y., Walters, W. W., Ni, Q., Liu, X., Xu, G., Shao, J., and Jiang, C.: Substantial nitrogen oxides emission reduction from China due to COVID-19 and its impact on surface ozone and aerosol pollution, Science of The Total Environment, 753, 142238, 2021.

Zheng, B., Geng, G. N., Ciais, P., Davis, S. J., Martin, R. V., Meng, J., Wu, N. N., Chevallier, F., Broquet, G.,

Boersma, F., van der Ronald, A., Lin, J. T., Guan, D. B., Lei, Y., He, K. B., and Zhang, Q.: Satellite-based estimates of decline and rebound in China's CO2 emissions during COVID-19 pandemic, Science Advances, 6, 2020.

---

## Author Response (AR1)

**Response to the reviewer 1**

The authors did an impressive work in collecting and analyzing activity data related to fossil fuel emissions during the COVID period. The results of this work can serve as a good reference in research related to the COVID impacts as well as atmospheric models. However, I believe that several issues could, and should, be further elaborated by the authors to help with the better understanding of this dataset.

We thank the reviewer for his/her interesting questions and comments. Please find below our answers.

1) line 109-112 says, "The collected data are then analyzed and an intercomparison of the changes in activity data from datasets providing similar or equivalent parameters are performed. The dataset that provides the most detailed and reliable data is then chosen."

This only applies to the transportation sector, correct? Because other sectors seem to suffer from lacking of data and the authors really didn't have much of a luxury to choose from different activity datasets for other sectors.

My question is, what is the standard for choosing the data in transportation sector? To be more specific, how did the authors define "reliable"?

The authors listed a comparison between the APPLE driving data and google mobility data. Figure S1 did show that APPLE data has more variations compared to google data. But does that mean APPLE data is less reliable? What is(are) the ultimate reason(s) for the discrepancies between APPLE data and google data? Besides the spatial coverage, what made the authors believe that google is more reliable than APPLE? Couldn't it be that in reality, the transportation emission IS that variant in different countries/regions?

For the road transport sector, we have indeed much more information than for the other sectors. For the industrial and aviation sectors (Figure S4 and S5, Supplementary material), two datasets were available. We are aware of the lack of up-to-date data on certain sectors of activity. This is why the manuscript indicates in lines 113-114: "an intercomparison of the changes in activity data of datasets providing similar or equivalent parameters is performed".

The use of the word "reliable" may not be the best word to use in this context. It has been replaced in the text by the term "which meets better our needs" (line 115). What we mean by our needs are the data that are publicly available for as many countries as possible, and where we can find information on a daily basis.

Our choice of Google instead of Apple related to the following analysis:

  - Google and Apple datasets do not measure the same mobility parameters: Apple bases their values on the volume of directions requests on phone applications while Google uses mobile phone locations. The comparison of these datasets indicates that the changes are similar when traffic is strongly reduced (March-May). However, from June to August, Apple data exhibit more variability compared to Google data, with a 2 to 3 times increase in mobility relative to the pre-covid period (Figure S4). These differences can be attributed to a combination of several factors including the spatial coverage, the mode and category of

transportation considered, the location of the measurements within the country or state/province. The calculation methods are also very different from one dataset to another. These two sentences have been added in the revised manuscript (lines 187-190).

      - Google provides a better spatial coverage. For example, Apple gives measurements for only 3 cities in Africa, while Google provides daily movement trends across 26 countries in Africa as indicated in lines 165-167.

2) Line 113-105, "The gridded daily/monthly files per sector are obtained by assigning the value of the AFs at the country/state/province level to each corresponding grid cell"

I have not found a detailed description on this in methodology. How exactly was the gridding implemented? The data sources listed in table 1 are mostly at national scale or province/state scale. How were they downscaled into 0.1 degree grids? What was used as the proxy? How reliable was the proxy?

This question is also addressed by the other reviewer, so our answer will also be included in the answers to the other reviewer.

In order to make the description of the methodology clearer, we have rephrased lines 117-122, and the new text is:

The gridded daily/monthly netcdf files developed for each sector are obtained by assigning, to each cell, the value of the AFs in the whole country or state/province level corresponding to this grid cell: this is based on the fact that the lockdowns and restrictions have been generally taken at national or states level. For several sectors (road transport, industry and residential) the AF at the country or state/province level represents an average value calculated from several individual cities or locations.

The figure below gives an illustration of the allocation of the AFs to each grid point at a 0.1x0.1 resolution in a country or a state/province.

[Figure]

3) Line 144-147,"In order to make the calculated AFs comparable with those derived using the other data sources considered in this study, the AFs for the Google's categories are scaled as a function of the Google mobility data, so that their values are less than 1 for a reduction in activity and above 1 otherwise."

I'm confused here. What is the baseline for 1? The authors mentioned in previous text in line 137-138 that "This baseline is calculated as the median value over the five-week period from January 3rd to February 6th 2020." But that was the Google's definition of baseline for their mobility data, not the same baseline used to calculate AF in this paper, correct?

Or was it the same thing? Please clarify.

Also, is the baseline the same for all other sectors? Please clarify.

To make the text clearer, we have changed lines 154-157, and the new text is the following: "In order to make the calculated AFs comparable with those derived using the other data sources considered in this study, the AFs for the Google's categories are scaled to 1 using the following formula AF = 1+ Google/100, so that their values are less than 1 for a reduction in activity and above 1 otherwise."

We use the same baseline period as Google, except for China for which the lockdowns started earlier. We use the same baseline for the other sectors, except for those for which we only have monthly activity data. For those sectors, we consider the value of January as the baseline. These details have been included in the revised manuscript (Lines 130-138).

4) This is not a question but rather a comment. The "uncertainty" as showed in Figure 2,3,4,5,6 is not the result of validation based on other reliable datasets, but instead, it refers to the AF variations at different geographical locations. In my opinion, the authors should call it as what it is instead of insinuating that it could be something that it is not. I'm not saying the

authors didn't do data validation, they did, as shown in figure S4 and S5. But the light pink color in figure 2,3,4,5,6 is not the result of validation.

The light pink color on Figures 2 to 6 shows indeed the standard deviation resulting from the AFs of all individual countries or states/provinces, as indicated in the legend of each figure. The text (lines 330-332) has been modified as follows: "The regional variations of AFs (light pink color) are determined as the standard deviation from individual values of all the countries in the region or from local measurements in the country or state/province."

**Response to the reviewer 2**

General comments:
Doumbia et al. present an interesting and important dataset on changes in global air pollutant emissions during the COVID-19 pandemic focusing on adjustment factors (AFs). This dataset is not only useful for global and regional emission inventories development, but also for atmospheric chemistry modeling, which is worth publishing in ESSD. I have several concerns on this paper.

We would like to thank the reviewer for agreeing to review our paper and for his/her valuable comments. Please find below our answers.

1. Concerning the spatial distribution, the Methodology does not provide enough details on how the 0.1x0.1 dataset was created, how to support the 0.1x0.1 resolution ? And the Results part showed only one spatial pattern figure (Fig. 8), which also need to be largely expanded not only for NOx.

This question is also addressed by the other reviewer, so our answer will also be included in the answers to the other reviewer.

In order to provide more details on the description of the methodology, we have rephrased lines 117-122, and the new text is:

"The gridded daily/monthly netcdf files developed for each sector are obtained by assigning, to each cell, the value of the AFs in the whole country or state/province level corresponding to this grid cell: this is based on the fact that the lockdowns and restrictions have been generally taken at national or state level. For several sectors (road transport, industry and residential) the AF at the country or state/province level represents an average value calculated from several individual cities or locations."

The figure below gives an illustration of the allocation of the AFs to each grid point at a 0.1x0.1 resolution in a country or a state/province.

[Figure]

Concerning your comment on the expansion of the spatial distribution to other species, we have added the following figure in the supplementary material (Figure S8), which shows the absolute changes in the total emissions of CO, NMVOCs, $SO_2$ and BC for April 2020.

[Figure]

2. Since the COVID-19 pandemic is still severe and the confirmed cases are increasing (https://www.worldometers.info/coronavirus/), this important dataset can aim to update it continuously, not only to the end of 2020 (lines 92-93).

We are indeed considering an update of the dataset until the end of the pandemics. For example, for the air traffic sector, we have recently extended the AFs to December 2020, as

shown in the figure below. These data will be included in the database during March 2021. We are compiling the available activity data for the other sectors, for a further update in the coming months.

[Figure]

3. How the AFs are calculated for different pollutants is not so clear with specific equations, especially when there are multiple sources activity data (e.g. industrial processes);

In our study, the estimated AFs are provided on a sector, geographic and temporal basis, not on a species basis. The AFs are provided for the sectors commonly used in emission inventories, i.e. energy, industry, residential and transportation. They can therefore be applied to any inventory providing the emissions of atmospheric pollutants.

4. Why are there larger decreases (30-50 %) in South America than other areas (e.g. China, EU and US) for NOx, NMVOCs and CO. And Fig.8 seems to show that China has the largest decrease, rather than South America ?

The larger decreases in South America shown in Figure 7 (Figure 8 in the revised manuscript) result from very strict restrictions in many countries in Latin America. For example, in Argentina, a very strict lockdown was implemented on March 19, 2020 and was extended several times until the end of the summer 2020. Figure 8 in the first version of the manuscript showed the spatial distribution of the absolute change in the Covid-19 NOx emissions compared to the reference scenario (without Covid-19) in China during the month of February while for the rest of the world the considered month is April, the periods of strictest restrictions.
To make these results clearer, we have added two figures in the supplement, Figure S7 and Figure S8, which show the absolute changes in total emissions of CO, NMVOCs, $SO_2$ and BC for April 2020 and the percentage changes at the global scale for the first six months of 2020, respectively.
We have also added a new figure (Figure 9 in the revised manuscript) showing the distribution of the absolute and percentage changes in NOx emissions for the month of April 2020.

Some minor comments:

1.The title might be too limited for only "atmospheric chemistry modeling";

The title has been changed to "Changes in global air pollutant emissions during the COVID-19 pandemic: a dataset for atmospheric modeling".

2. Line 51 are repeated for CO with lines 48-49;

We have removed CO from the sentence.

3. Line 65 for greenhouse gas, there are other references (Han et al., 2021; Zheng et al., 2020);

These references have been included, in line 67 of the revised manuscript.

4. Line 81-82, need to expand the meanings of this dataset ? e.g. provide assessments on COVID-19 restrictions on pollutant emissions;

This remark is considered in the revised manuscript. The sentence, in line 82-85 of the revised manuscript is rewritten as follows: "The advantage of such a dataset, which provides adjustment factors for assessments of the impact of COVID-19 restrictions on pollutant emitted into the atmosphere, is that it can be applied directly to any global or regional inventory used in chemistry-climate and transport models in a flexible way".

5. Line 95, could $CO_2$ be included? It would be better for homology studies.

Though the AFs from the CONFORM dataset can be applied to the emissions of any species, we have only considered in section 3.5 its application to the emissions of the species indicated in line 98 of the revised manuscript.

6. Lines 116-117: Add some introduction on CAMS -GLOB-ANT;

We have provided the following information on the CAMS-GLOB-ANT dataset in section 3.5, lines 449-452: "This dataset (CAMS-GLOB-ANT) provides daily emissions of the main atmospheric compounds, including speciated volatile organic compounds at a 0.1°x0.1° resolution, from 2000 to 2020. Version R.1 of the CAMS-GLOB-ANT_v4.2 dataset incorporates the MEIC1.3 regional inventory for China described by Zheng et al. (2018)."

7. Lines 119-120 repeated with Lines 92-93;

Lines 119-120 have been removed.

8. Line 157: In reference to (Han et al., 2021), monthly road (and also ship) transportation data for China can be obtained at http://www.mot.gov.cn/tongjishuju/, but it needs some translation and digitalization work to obtain the data. And it is up to the authors to decide whether to include such data;

We thank the reviewer for providing this information. We might consider this dataset in another study, where we could compare our dataset obtained from Baidu (https://qianxi.baidu.com/) with this dataset.

9. Lines 197-198: Maybe need some discussions. The assumption is not very consistent with Liu et al., 2020, see (https://www.carbonmonitor.org.cn/user/data.php?by=cn);

[Figure]

We thank the reviewer for provided this information from the carbon monitor website. The data proposed by the reviewer provide emissions of $CO_2$ for road transport in China from January to November: we retrieved the data and found an average difference between 2020 and 2019 of about 4% for the period from May to November, with larger values for some days (see figure below).

[Figure]

We compared the changes in the Baidu Migration Scale Index with the relative difference of TomTom congestion levels for the Beijing, Tianjin, Chongqing and Shanghai Chinese areas. TomTom archived data are not freely available, but they can be retrieved from published graphs (https://www.tomtom.com/en_gb/traffic-index/ranking) providing the weekly changes in 2020 relative to the same periods in 2019. The results show a strong similarity between changes given by these two datasets for the period covered by the Baidu dataset (January to April), with a correlation coefficient of 0.9 (Figure S3). For the period from May to August 2020, Liu et al. (2020) showed that the difference in estimated $CO_2$ emissions from road transport between 2020 and 2019 is on average about 4% for May-November, with larger values for some days. Based on these analyses, and in order to cover the whole period of our study (January to August), we assume that changes in road traffic in China after May 2020 are relatively low and close to those before the spread of the COVID-19 pandemic. These comparisons show that the proposed method for calculating the AFs (i.e. ratio between the activity data and the median value of activity data over a defined reference period) is consistent with changes in 2020 relative to the same period in 2019.

10. Lines 201-202: add a reference;

The link for accessing to the TomTom congestion data has been added to the revised manuscript (Line 211).

11. Line 205: May need to add time resolutions (e.g. daily or monthly) for sectors in Table 1;

The time resolutions for the different sectors considered in this dataset are now added in Table 1.

12. Line 211: Cement is not included in this dataset?

Cement is not explicitly included in our dataset. Industrial activities are considered as whole, using the Google's workplaces measurements.

The indication of cement was just an illustration of what could be considered in this sector as we defined it, since its definition can be different from one inventory to another. To avoid a confusion from the readers, we have rewritten the sentence as (lines 226-227):

"This sector includes industrial production processes such as manufactured products from fossil fuel combustion, and represents a significant part of the emission sources of atmospheric pollutants".

13. Line 216: For China monthly data, the iron, steel and cement production can be obtained at https://data.stats.gov.cn/english/easyquery.htm?cn=A01 in "Output of major industrial products"

We have already checked that the trend of steel production from China's national statistic is similar to those derived from the World steel as used in the paper. For the industrial sector, the AFs are estimated from Google's workplaces category. In China we assume that coal consumption from the six main coal producers is representative of changes over this sector. Monthly steel productions are therefore used only for evaluating the use of Google mobility instead of monthly national data (Figure S4, Supplementary information).

We thank the reviewer for proposing a site with data for China. However, the site indicated by the reviewer could not be reached by the co-authors working in Europe. We tried 3 different browsers on computers connected to the internet through 4 different providers, and the answer was always the same, i.e. "Warning – Potential Security Risk: a security threat was detected". After contacting the system administrators of our institutes, we were told that we should not try to connect to this site, and that we cannot refer to it in any paper.

14. For Section 2.3, China's power data can be reflected by daily coal consumption at six main power groups, see (Han et al., 2021) Fig.4, and the data is provided at the end of this file.

The AFs for the power sector in China was estimated based on emission data published by Forster et al., 2020. At the time we were writing the paper, this was the only data we could access to. However, the comparison between Forster et al., 2020 and Han et al., 2021 data shows rather similar AF trends as indicated in the figure below. As indicated above, the reference to Han et al. (2021) has been added to the manuscript.

[Figure]

15. Lines 251-252: Can compare with data from International Civil Aviation

Organization (ICAO). I noticed a report which contained regional/country data (https://www.icao.int/sustainability/Documents/COVID-19/ICAO_Coronavirus_Econ_Impact.pdf), which you may find useful in cross validation;

[Figure]

We have used the data from organizations provided the number of flights operating during the year 2020. The dataset recommended by the reviewer does not provide data as accurate as the ones we have used: ICAO provides the number of passengers carried in 2020, compared to 2019. However, as the number of passengers carried does not reflect the number of flights operating, we have preferred to use air traffic data, as the number of planes flying is a better indication of the emissions.

16. Line 324: Also consider these refs. (He et al., 2021 ; Sun et al., 2020; Zhang et al., 2021);

These references have now been included in the paper, in lines 339 and 340.

17. Lines 375-377: May be not this case (see below 3 figures), and CO2 emissions from Liu et al., (2020) (https://www.carbonmonitor.org.cn/user/data.php?by=cn) and NBS statistical

data on iron and steel and cement productions
(https://data.stats.gov.cn/english/easyquery.htm?cn=A01) showed that industry in China recovered soon after April 1st, and surpassed the before COVID-19 mean state.

[Figure]

[Figure]

[Figure]

Thank you for this helpful information. For the industrial sector, we extracted the coal consumption data for the six major firms in China from Myllyvirta 2020 (https://www.carbonbrief.org/analysis-coronavirus-has-temporarily-reduced-chinas-co2-emissions-by-a-quarter). The comparison with data from Liu et al., 2020 shows clear discrepancies from May to August. Note that, at the time we were retrieving the data from the carbonbrief website, emissions were available only until May 2, 2020 and we extrapolated them until August. This might explain the underestimation of the increase in the industrial activity in China after April. In the revised manuscript, the AFs for such sector have been updated using the data published by Liu et al., 2020 (see Figure below). The reference provided by the reviewer is added in Table 1. We have updated Figure 4 as well as the

CONFORM dataset and the text has changed in lines 392-394 as follow: "In China, AF fell in mid-February to its minimum average value of 0.60 (40 % decrease), but rapidly increased to complete recovery at the beginning of March and exceeded the pre-pandemic level by an average 25 % from April onward".

[Figure]

18. Line 406: Should be Figure 6 for power and Figure 4 for Industry?

Yes, Figure 4 is for the industry sector and Figure 6 for the power as indicated in the legend of the figures. The correct number has been included in the text.

19. Lines 411-412: Power AFs for China is not consistent with (Liu et al., 2020) and (Han et al., 2021), I provided the daily coal consumption data for six major power generation groups at the end of minor comments for your reference.

As already mentioned in our response to comment 14, the AFs for the power sector in China were estimated on the basis on emission data published by Forster et al., 2020. At the time we were writing the paper, this was the only data we had access to. However, the comparison between the Forster et al., 2020 and Han et al., 2021 data shows rather similar AF trends as indicated in the figure below.

[Figure]

20. Line 426: change the red color "is" to black;

We have changed the text accordingly.

21. Lines 449-450: NMVOCs are mainly from solvents and industrial processes (Lines 464-465), and not homogeneous with SO2? Here "rather similar" seems to show some relations?

There are no relations between the emissions of $SO_2$ and NMVOCs. In order to make the sentence clearer, we have rephrased lines 474-475 to:

The NMVOCs emissions decrease significantly, by 22 % (15-29 %). The decreases in the amount of $SO_2$ emissions are of the same order of magnitude.

22. Lines 478-480: Expand this short paragraph a bit;

Important modifications have been performed in section 3.5 to take into account the reviewer's comments 22 and 23.

23. Line 486: "largest" or "large"?

The text has been changed in the section corresponding to the reviewer's comments.

24. Line 513: "might" or "could" or "did"?

The right word is "could". The text has been changed accordingly.

25. Draw vertical lines to show lockdown and unlock date information in time series figures for major countries or regions.

We have considered the suggestion of the reviewer, showing the dates of the lockdowns for the major countries. However, we think such a figure would be rather misleading: the lockdowns implemented in the different countries of the world were very different. Some were very strict, some were not. Some publications have tried to provide such information, and we have included this information in the supplement (page 1), such as:

https://en.wikipedia.org/wiki/COVID-19_lockdowns

https://www.bbc.com/news/world-52103747
https://www.statista.com/chart/22048/university-of-oxford-coronavirus-containment-and-health-index-selected-countries/

If new websites providing information on 2020 and 2021 lockdowns become available, more links will be added to this section of the supplement.

References added to the manuscript:
Han, P. F., Cai, Q. X., Oda, T., Zeng, N., Shan, Y. L., Lin, X. H., and Liu, D.: Assessing the recent impact of COVID-19 on carbon emissions from China using domestic economic data, Science of the Total Environment, 750, 2021.

He, Y., Pan, Y., Gu, M., Sun, Q., Zhang, Q., Zhang, R., and Wang, Y.: Changes of Ammonia Concentrations in Wintertime on the North China Plain from 2018 to 2020, Atmospheric Research, doi: https://doi.org/10.1016/j.atmosres.2021.105490, 2021. 105490, 2021.

Liu, Z., Ciais, P., Deng, Z., Lei, R., Davis, S. J., Feng, S., Zheng, B., Cui, D., Dou, X., Zhu, B., Guo, R., Ke, P., Sun, T., Lu, C., He, P., Wang, Y., Yue, X., Wang, Y., Lei, Y., Zhou, H., Cai, Z., Wu, Y., Guo, R., Han, T., Xue, J., Boucher, O., Boucher, E., Chevallier, F., Tanaka, K., Wei, Y., Zhong, H., Kang, C., Zhang, N., Chen, B., Xi, F., Liu, M., Bréon, F.-M., Lu, Y., Zhang, Q., Guan, D., Gong, P., Kammen, D. M., He, K., and Schellnhuber, H. J.: Near-real-time monitoring of global CO2 emissions reveals the effects of the COVID-19 pandemic, Nature Communications, 11, 5172, 2020.

Sun, Y., Lei, L., Zhou, W., Chen, C., He, Y., Sun, J., Li, Z., Xu, W., Wang, Q., Ji, D., Fu, P., Wang, Z., and Worsnop, R. D.: A chemical cocktail during the COVID-19 outbreak in Beijing, China: Insights from six-year aerosol particle composition measurements during the Chinese New Year holiday, ence of The Total Environment, 742, 2020.

Zhang, Q., Pan, Y., He, Y., Walters, W. W., Ni, Q., Liu, X., Xu, G., Shao, J., and Jiang, C.: Substantial nitrogen oxides emission reduction from China due to COVID-19 and its impact on surface ozone and aerosol pollution, Science of The Total Environment, 753, 142238, 2021.

Zheng, B., Geng, G. N., Ciais, P., Davis, S. J., Martin, R. V., Meng, J., Wu, N. N., Chevallier, F., Broquet, G.,
Boersma, F., van der Ronald, A., Lin, J. T., Guan, D. B., Lei, Y., He, K. B., and Zhang, Q.: Satellite-based estimates of decline and rebound in China's CO2 emissions during COVID-19 pandemic, Science Advances, 6, 2020.

---

## Author Response (AR2)

**Response to the Referee #3**

We thank the reviewer for his/her interesting comments. The answers to the reviewer's questions are highlighted in red below.

This study provides spatially resolved sectorial adjustment factors (AFs) of emissions during the COVID lockdown. As pointing out by the authors, this database is expected to be directly applied in emission changes which can be further used in global or regional inventories in air quality modelling.
While this database is useful and desires for a publication in ESSD, I think it is necessary and helpful to clarify:

1) For road transport, while the information from Google or Baidu maps provides generally the transport intensities on road, emissions from different vehicle types and gasoline/diesel are different. Were this considered in the emission AFs ?

As a general comment, the CONFORM dataset was designed to be directly applicable to existing global and regional inventories (those used by chemical transport models) that use relatively similar sectors (EDGAR, ECLIPSE, CAMS, MEIC, REAS, etc.). These inventories include emissions from the following six major sectors: industrial processes, road transportation, power generation, residential, aviation and shipping. As stated in the paper, we developed adjustment factors for the sectors used in these or similar inventories; however, these inventories do not provide information on emissions by vehicle or fuel type. It is worth noting that our approach is similar to that used for greenhouse gases (e.g. Le Quéré et al., 2020).

To clarify this point raised by the reviewer in the manuscript, we have added the following text to the introduction section: " The inventories commonly used in these models (EDGAR (Crippa et al., 2018), ECLIPSE (Klimont et al., 2017), CAMS (Granier et al., 2019), MEIC (Li et al., 2017), etc.) include the sectors already mentioned (industrial processes, road transportation, power generation, residential, aviation and shipping). The emissions of these sectors are developed on the basis of a combination of several sub-sectors that are not provided separately in the emission inventories." (Lines 84-89).

2) Industrial sector- workplace change might be an indicator, as it is hard to get accurate and reliable data on this. But this estimation is expected to have much higher uncertainties, and the steel production activities are rather different from many others, for example, petroleum industry facilities most of which are not shut down during the COVID.

We highlighted the uncertainty in the estimation of AFs for industrial sector using Google's workplace category in lines 242-245 such as "… there are large differences in some countries between these data. For example, in Europe the greatest change in crude steel production is 24 % in comparison to 59 % estimated using Google's workplace category, indicating a high level of uncertainty in the AFs for the industry sector". The difference between the maximum AF estimated from steel production and Google's workplace is 35 %, which is within an order of magnitude of the maximum uncertainty estimated by Liu at al. (2020) for this sector of 36 %.
Furthermore, we mentioned in lines 398-399 of the manuscript that the AFs for the industrial processes sector are subject to average uncertainties ranging from ±20 to ±30 % depending on the regions.

3) For power plants, were data for electricity plants using different fuels- coal-fired, nuclear power, hydroelectric power etc.,

For the estimation of emission AFs in the power sector, we use total electricity load activity data from various fuels such as coal, gas, nuclear and hydroelectric power. However, as previously stated, the majority of current emission inventories commonly used for atmospheric modeling do not include emissions for these specific sub-sectors (coal, gas, nuclear power, hydroelectric power, etc.). As a result, the reviewer's requested level of detail is unavailable and cannot be applied to the inventories targeted by the CONFORM dataset.

4) Residential sector-I agreed with the authors that there was an increase in residential emissions, which could be also found in some recent studies finding high indoor air pollution (also leading to higher overall exposure) during the COVID in rural area. But in urban homes, the increased electricity contributed small to the increase of emissions (in fact in most emission inventories, residential electricity associated emissions are not counted in the residential sector), and gas burning for cooking increased very small. People had three meals per day, no matter it is during the COVID or not. However, the AFs based on the increased electricity data for London are not representative for other countries using solid fuels. Why not referring to information from other developing countries, especially those using solid fuels in rural area ?

Changes in emissions from the residential sector are difficult to assess across the different regions of the globe due to differences in the exact definition of the sector from one inventory to the next. For example, in Le Quéré et al. (2020), $CO_2$ emissions from the residential sector were estimated using a combination of confinement indexes and electricity consumption of the city of London, which was then extrapolated to the other countries. Liu et al. (2020) used fuel consumption data from 2019 that was scaled to 2020 based on the population-weighted heating degree days variation between these two years. Although a more detailed study based on national economic activity data that are not yet available in many countries could be conducted in the future. In this study, we used Google mobility reports (https://www.google.com/covid19/mobility/) and assumed that emission adjustment factors using Google's residential category are representative of the majority of domestic combustion activities (cooking, residential heating, heating water and other combustion activities) in many countries. We only used residential emissions from Le Quéré et al. (2020) to derive adjustment factors for China, as stated in lines 318-319 of the revised manuscript.

5) Besides changes in individual residential homes, there are significant changes in commercial emissions for example restaurants and the mall. Was this considered and available from some data ?

The residential sector is defined in the CONFORM dataset in the same way that it is defined in the EDGAR emission dataset (Crippa et al., 2018), namely as a sector that includes both residential and commercial activities. Almost all inventories lack precise information about the residential sector, and estimates of the residential and commercial sectors' contributions to emissions are currently unavailable. As stated in the previous answer, the lack of detailed information for estimating adjustment factors for the residential sector results in a high level of uncertainty for this sector, which is approximately 20 % in our study and approximately 40 % in the study of Liu et al., 2020, as indicated in the manuscript (Lines 416-419).

6) Emissions are different from different fuels- fossil and biomass ones. This is more obvious in residential sector, where multiple different fuels are used. Residential coal and biomass use contribute largely to the primary emissions of PM2.5, BC and OC. Were different AFs for different fuels types, and differences in pollutant species, considered in the development of database ?

As previously stated, the AFs are provided for a number of sectors that are commonly used in current emission inventories, such as energy, industry, residential, road transportation, aviation and shipping. Almost all of the inventories listed here and generally used in atmospheric models lack information on emissions by fuel type. AFs are calculated in our study on a sector, geographic, and temporal basis rather than by pollutant species.

7) Validation of results is always an important concern. It is accepted that global or regional emission inventory itself is difficult to be validated unless in couple with the air transport chemical models and validated in comparison to the monitoring data. This should be discussed in the manuscript, if it is presently not in the study scope.

The J. of Geophys. Res. has published an evaluation of our dataset (Gaubert et al., 2021). Both our current ESSD paper and the J. Geophys. Res. study were submitted in November 2020. Gaubert et al. (2021) performed simulations with the global Community Atmosphere Model (CAM-Chem), after applying the CONFORM dataset to the CAMS-GLOB-ANT_v4.2-R1.1 anthropogenic emissions. The figure below depicts the percentage change in the concentrations of many chemical compounds in China as a result of reduced primary pollutant emissions in February 2020 during the COVID-19 pandemic (Figure 6 in the J. Geophys. Res. publication). The figure shows that during the strict lockdown, the surface concentration of NOx was significantly reduced (40–50%) in most areas of eastern China and the country's northwest. At the same time, as evidenced by surface observations, ozone concentrations increased throughout the northeastern part of China and locally in a number of significant urban areas in other regions (e.g., Huang et al., 2020; Shi & Brasseur, 2020). In addition, there was a decrease in ozone in the southern part of the country. This finding is consistent with Liu and Wang's (2020) regional model analysis and surface observations (e.g., Fu, Wang, et al., 2020; Lian et al., 2020).

[Figure]

Another assessment of our dataset in China was published in the journal Science of the Total Environment (Liu et al., 2021). Using observed and predicted data, Liu et al. (2021) investigate the surface ozone before and during the lockdown. The CONFORM dataset was used in the CMAQ (Community Multiscale Air Quality model, v5.2.1) model (US EPA, 2018) after being applied to the MEIC (Multi-resolution Emission Inventory for China) regional emission inventory (Li et al., 2017). The findings show that reductions in anthropogenic emissions of ozone precursors (NOx and VOCs) contributed to changes in surface ozone that are consistent with observations. This newly published study has been cited in the revised manuscript (Lines: 450:454).

Bouarar et al. (2021), a paper submitted to Geophysical Research Letters, use the CAM-Chem model to evaluate the performance of CONFORM dataset by simulating the response of chemical species in the free troposphere. Another paper, based on TROPOMI and IASI satellite measurements and model simulations, was also recently submitted to GRL, and it investigated the impact of COVID-19 on NOx and VOCs chemical compounds over China. According to both studies, model simulations using anthropogenic emissions and the CONFORM dataset capture the observed variations in ozone concentrations in the free troposphere during the Northern Hemisphere spring/summer, as well as in spaceborne observations of $NO_2$ and VOCs in China.

References:

Crippa, M., Guizzardi, D., Muntean, M., Schaaf, E., Dentener, F., van Aardenne, J. A., Monni, S., Doering, U., Olivier, J. G. J., Pagliari, V., and Janssens-Maenhout, G.: Gridded emissions of air pollutants for the period 1970–2012 within EDGAR v4.3.2, Earth Syst. Sci. Data, 10, 1987–2013, https://doi.org/10.5194/essd-10-1987-2018, 2018.

Forster, P. M., H. I. Forster, M. J. Evans, M. J. Gidden, C. D. Jones, C. A. Keller, R. D. Lamboll, C. Le Quéré, J. Rogelj, D. Rosen, C.-F. Schleussner, T. B. Richardson, C. J. Smith and S. T. Turnock: Current and future global climate impacts resulting from COVID-19, *Nat. Clim. Chan.*, 10, 913-919, doi:10.1038/s41558-020-0883-0, 2020.

Gaubert, B., Bouarar, I., Doumbia, T., Liu, Y., Stavrakou, T., Deroubaix, A., et al. (2021). Global changes in secondary atmospheric pollutants during the 2020 COVID-19 pandemic. Journal of Geophysical Research: Atmospheres, 126, e2020JD034213. https://doi.org/10.1029/2020JD034213.

Bouarar, I., B. Gaubert, G. P. Brasseur, W. Steinbrecht, T. Doumbia, S. Tilmes, Y. Liu, T. Stavrakou, A. M. Deroubaix, S. Darras, C. Granier, F. G. Lacey, J-F. Müller, X. Shi, N. Elguindi and T. Wang : Ozone Anomalies in the Free Troposphere during the COVID-19 Pandemic, 10.1002/essoar.10506600.1, 2021.

Fu, X., Wang, T., Gao, J., Wang, P., Liu, Y., Wang, S., et al. (2020). Persistent heavy winter nitrate pollution driven by increased photochemical oxidants in northern China. Environmental Science & Technology, 54(7), 3881– 3889. https://doi.org/10.1021/acs.est.9b07248.

Huang, X., Ding, A., Gao, J., Zheng, B., Zhou, D., Qi, X., & He, K. (2020). Enhanced secondary pollution offset reduction of primary emissions during COVID-19 lockdown in China. National Science Review, 8, nwaa137. https://doi.org/10.1093/nsr/nwaa137.

Le Quéré, C., R. B. Jackson, M. W. Jones, A. J. P. Smith, S. Abernethy, R. M. Andrew, A. J. De-Gol, D. R. Willis, Y. Shan, J. G. Canadell, P. Friedlingstein, F. Creutzig and G. P. Peters: Temporary reduction in daily global CO2 emissions during the COVID-19 forced confinement, Nat. Clim. Chang. 10, 647–665, doi:10.1038/s41558-020-0797-x, 2020.

Li, M., Liu, H., Geng, G., Hong, C., Liu, F., Song, Y., Tong, D., Zheng, B., Cui, H., Man, H., Zhang, Q., and He, K.: Anthropogenic emission inventories in China: a review, Natl. Sci. Rev., 4, 834-866, doi: 10.1093/nsr/nwx150, 2017.

Lian, X., Huang, J., Huang, R., Liu, C., Wang, L., & Zhang, T. (2020). Impact of city lockdown on the air quality of COVID-19-hit of Wuhan city. Science of the Total Environment, 742, 140556. https://doi.org/10.1016/j.scitotenv.2020.140556.

Liu, Y.M., Wang, T., Stavrakou, T., Elguindi, N., Doumbia, T., Granier, C., Bouarar, I., Gaubert, B., Brasseur, G.P. Diverse response of surface ozone to COVID-19 lockdown in China. Science of the Total Environment. https://doi.org/10.1016/j.scitotenv.2021.147739 (In Press).

Liu, Z., Ciais, P., Deng, Z., Lei, R., Davis, S. J., Feng, S., Zheng, B., Cui, D., Dou, X., Zhu, B., Guo, R., Ke, P., Sun, T., Lu, C., He, P., Wang, Y., Yue, X., Wang, Y., Lei, Y., Zhou, H., Cai, Z., Wu, Y., Guo, R., Han, T., Xue, J., Boucher, O., Boucher, E., Chevallier, F., Tanaka, K., Wei, Y., Zhong, H., Kang, C., Zhang, N., Chen, B., Xi, F., Liu, M., Bréon, F-M., Lu, Y., Zhang, Q., Guan, D., Gong, P., Kammen, D. M.,  He, K., Schellnhuber, H. J.: Near-real-time monitoring of global

$CO_2$ emissions reveals the effects of the COVID-19 pandemic, Nat. Commun. 11, 5172, https://doi.org/10.1038/s41467-020-18922-7, 2020.

Shi, X., & Brasseur, G. P. (2020). The response in air quality to the reduction of Chinese economic activities during the COVID-19 outbreak. Geophysical Research Letters, 47, e2020GL088070. https://doi.org/10.1029/2020gl088070.

Stavrakou, T., J.-F. Müller, M. Bauwens, T. Doumbia, N. Elguindi, S. Darras, C. Granier, I. De Smedt, C. Lerot, M. Van Roozendael, B. Franco, L. Clarisse, C. Clerbaux, P.-F. Coheur, Y. Liu, T. Wang, I. Bouarar, X. Shi, B. Gaubert, S. Tilmes, G. Brasseur: Impact of COVID-19 on NOx and VOC compounds over China based on TROPOMI and IASI satellite observations and model simulations. Submitted to Geophysical Research Letters, 2021.

United States Environmental Protection Agency. (2018). CMAQ (Version 5.2.1) [Software]. Available from https:// doi:10.5281/zenodo.1212601.

---

## Author Response (AR3)

Response to the Editor's comment

Dear editor,

Thank you for your email regarding our paper essd-2020-348. The Topical editor proposes that air traffic data published in the essd-13-357-2021 paper be considered. The OpenSky Network used in the essd-13-357-2021 paper, as stated in the revised manuscript (lines 291-295), provides similar trends of flight activities as the Official Aviation Guide (OAG), which we used in conjunction with the KCDM dataset (see attached plot showing the comparaison between OpenSky Network and OAG data for some regions). The main difference between the two datasets is the time step for which they are available. The reference to the paper essd-13-357-2021 has been added to the manuscript.

We also stated in the revised manuscript that the CONFORM dataset is now available until December 2020 (line 578).

We also made a minor change to the text on lines 457-463 to include a recently published paper on the evaluation of CONFORM data.

All these changes we have included in the submitted revised manuscript.

We hope that the changes in the manuscript will answer your comments.

Sincerely,

Thierno Doumbia

[Figure]

Weekly Air Traffic change in 2020 relative to 2019